# Tight analyses of first-order methods with error feedback

**Daniel Berg Thomsen**[1,2*]  **Adrien Taylor**[1]  **Aymeric Dieuleveut**[2]

[1]INRIA, D.I. École Normale Supérieure, PSL Research University, 75005 Paris, France
[2]CMAP, CNRS, École polytechnique, Institut Polytechnique de Paris, 91120 Palaiseau, France

## Abstract

Communication between agents often constitutes a major computational bottleneck in distributed learning. One of the most common mitigation strategies is to compress the information exchanged, thereby reducing communication overhead. To counteract the degradation in convergence associated with compressed communication, error feedback schemes—most notably EF and $EF^{21}$—were introduced. In this work, we provide a *tight analysis* of both of these methods. Specifically, we find the Lyapunov function that yields the best possible convergence rate for each method—with matching lower bounds. This principled approach yields sharp performance guarantees and enables a rigorous, apples-to-apples comparison between EF, $EF^{21}$, and compressed gradient descent. Our analysis is carried out in the simplified single-agent setting, which allows for clean theoretical insights and fair comparison of the underlying mechanisms.

> **Remark: proof certificates**
>
> To consolidate and support our theoretical results, we complement each theoretical statement with analytical or numerical validation. Specifically, we provide certificates of correctness generated either with a *Computer Algebra System* (CAS), using a WolframScript, for symbolic verification, or using *Performance Estimation Problems (PEP)* for numerical validation. CAS enable verification of algebraic identities, while PEP annotations indicate numerical validation of complete statements. These certificates are highlighted in the paper using [CAS] and [PEP] markers, which are direct links to the corresponding Jupyter notebook or WolframScript in our public GitHub repository. [a]

## 1  Introduction

Over the past decade, distributed optimization has become a cornerstone of large-scale machine learning. This shift is driven by major increases in the size of models and training data, as well as increasing societal concerns about data ownership and privacy. Ultimately, solutions in which training is distributed across a network of $n$ agents, each retaining its own local data, under the coordination of a central server, have emerged as one of the most natural and efficient solutions to this problem [1, 2].

---

*Correspondence to daniel.berg-thomsen@inria.fr

[a]While these certificates do not replace the mathematical proofs presented in the paper, they serve as an additional layer of transparency and error checking, analogous to unit tests in software development. This practice provides a reproducible, independently verifiable basis for our theoretical claims, thereby reducing the risk of oversights in complex derivations.

39th Conference on Neural Information Processing Systems (NeurIPS 2025).

Formally, the goal is to solve the following minimization problem:

$$\min_{x \in \mathbb{R}^d} \left\{ f(x) := \frac{1}{n} \sum_{i=1}^{n} f_i(x) \right\}. \tag{1}$$

Classical methods such as distributed gradient descent and its stochastic variants achieve linear speedups in iteration complexity with respect to the number of agents. However, they often suffer from significant *communication overhead*, as gradients or model updates must be exchanged frequently over bandwidth-limited channels [3–5]. As the scale of models keep increasing, this communication bottleneck has been identified early on as a critical limitation, prompting the development of methods aimed at reducing communication costs. Two main strategies are favored: scarcely communicating with the central server, known as *local iterations* [see e.g. 1, 6]—and transmitting *compressed updates*, which aim to reduce the size of the exchanged information. Compression mechanisms can be applied to reduce communication either from agents to the server [3, 7–14] or from the server to the agents [15–23]. This paper focuses on methods using compression operators, which encompass a variety of strategies, including selecting only a fraction of the weights to be transmitted (e.g., the top $K$ coordinates [8]) or communicating low-precision updates via quantization [7].

---

**Algorithm 1** Compressed gradient descent (CGD)

---

1: **initialization:** $x_0 \in \mathbb{R}^d, \eta > 0$
2: **for** $k = 0, 1, 2, \ldots, N$ **do**
3:     Agent $i \in [n]$ compresses $\nabla f_i(x_k)$ and communicates $m_k^{(i)} := \mathcal{C}(\nabla f_i(x_k))$
4:     Server updates $x_{k+1} \leftarrow x_k - \eta \cdot \frac{1}{n} \sum_{i=1}^{n} m_k^{(i)}$
5: **end for**

---

Formally, a compression operator is a possibly random mapping $\mathcal{C} : \mathcal{X} \to \mathcal{X}$, such that $\mathcal{C}(X)$ can be encoded (almost surely or on average) with a lower number of bits than $X$. The most natural algorithm leveraging communication compression with a centralized server is the *compressed gradient descent* algorithm (CGD), which is described in Algorithm 1. The main idea is to perform a distributed gradient step, with the compression operator $\mathcal{C}$ applied to the gradient of each agent before communication. Although this compression scheme reduces the communication cost, it comes at the expense of non-convergence in any practical setting [24].

To assess the general impact of compression schemes on the rate of convergence, one typically leverages the fact that these compressors all satisfy generic assumptions. These include unbiasedness, i.e., $\mathbb{E}[\mathcal{C}(x)] = x$ for any $x \in \mathcal{X}$, together with relatively bounded variance, which states that $\mathbb{E}[\|\mathcal{C}(x) - x\|^2] \leq \omega \|x\|^2$ for any $x \in \mathcal{X}$ [7, 11, 9, 25–27, 12, 28, 19, 20, 29–31], or *contractiveness* [32–34, 21, 24], defined as follows:

**Assumption 1** (Contractive compression operator). *The compression operator $\mathcal{C}$ is a stochastic operator such that, for some $\epsilon \in [0, 1)$,*

$$\textit{for all} \quad x \in \mathbb{R}^d, \quad \mathbb{E}\left[\|x - \mathcal{C}(x)\|^2\right] \leqslant \epsilon \|x\|^2. \tag{2}$$

The standard way to improve CGD is to leverage the asymmetry of information: each agent has access to the exact gradient before compression and can therefore track the discrepancy between the exact gradient and the transmitted (compressed) message. This discrepancy can be stored and used as a correction term in subsequent iterations—a principle that lies at the heart of *error feedback techniques*. The most basic mechanism used is known as *classic error feedback* (EF), where each agent stores the difference between the true gradient and its compressed version locally, and incorporates this error into the next round of communication. This method, outlined in Algorithm 2, was first introduced in [3] and later analyzed in [32, 11, 10, 8, 35]. Notably, this method converges in many practical settings, effectively addressing the problem of non-convergence for CGD.

More recently, a variant of the classic error feedback mechanism, known as $\text{EF}^{21}$ was introduced by Richtárik et al. [14], and is presented in Algorithm 3. Unlike classic error feedback, $\text{EF}^{21}$ focuses on communicating a gradient estimate that is more robust to the variance observed in gradients received from *different* agents around the minimum of finite sum objectives. This method has since been extended in several directions [e.g. 23, 36–38].

---

**Algorithm 2** Classic error feedback (EF)

---

1: **initialization:** $x_0 \in \mathbb{R}^d, \eta > 0, e_0^{(i)} = 0$ for $i = 1, \ldots, n$
2: **for** $k = 0, 1, 2, \ldots, N$ **do**
3:      Agent $i \in [n]$ compresses $e_k^{(i)} + \eta \nabla f_i(x_k)$ and communicates $m_k^{(i)} := \mathcal{C}(e_k^{(i)} + \eta \nabla f_i(x_k))$
4:      Agent $i \in [n]$ updates $e_k^{(i)} \leftarrow e_k^{(i)} + \eta \nabla f_i(x_k) - \mathcal{C}(e_k^{(i)} + \eta \nabla f_i(x_k))$
5:      Server updates $x_{k+1} \leftarrow x_k - \frac{1}{n} \sum_{i=1}^n m_k^{(i)}$
6: **end for**

---

---

**Algorithm 3** Error Feedback 21 — EF$^{21}$

---

1: **initialization:** $x_0 \in \mathbb{R}^d$; step size $\eta > 0$; $d_0^{(i)} = \mathcal{C}(\nabla f_i(x_0))$ for $i = 1, \ldots, n$;
2: **for** $k = 0, 1, 2, \ldots, N$ **do**
3:      Server updates $x_{k+1} \leftarrow x_k - \eta \cdot \frac{1}{n} \sum_{i=1}^n d_k^{(i)}$
4:      Agent $i \in [n]$ compresses $\nabla f_i(x_{k+1}) - d_k^{(i)}$ and communicates $m_k^{(i)} := \mathcal{C}(\nabla f_i(x_{k+1}) - d_k^{(i)})$
5:      Agent $i \in [n]$ updates $d_{k+1}^{(i)} \leftarrow d_k^{(i)} + m_k^{(i)}$
6: **end for**

---

Error feedback techniques are widely regarded as highly effective, and EF was described as *"compression for free"* as early as 2019 [10]. Despite that, and the abundant literature on the topic, the precise impact of error feedback techniques on performance remains difficult to assess. Comparison is complicated by the diversity of settings under which methods are analyzed: different function classes (smooth, convex, or nonconvex), a range of algorithmic enhancements (acceleration, adaptivity, variance reduction, etc.), and a variety of performance measures (different Lyapunov functions) [23, 39, 38, 37, 40–42]. While some works provide insightful counter-examples—e.g., Beznosikov et al. [24] show that classic error feedback effectively addresses the limitations of CGD in distributed settings—many others simply propose a Lyapunov function and establish an upper bound without demonstrating its tightness. As a result, claims about "compression for free" are often based on comparisons between potentially loose guarantees, which may not reliably reflect real algorithmic performance. The length and complexity of the proofs involved typically make it difficult to ensure the tightness of the results, and most proofs are constructed in an ad hoc manner. Consequently, it is difficult to determine which methods are actually worst-case optimal based on upper bounds whose tightness is not always assured.

As a result, even remarkably simple questions remain only partially answered:

> What is the optimal convergence rate that each method can attain?
> Given an optimization setting, what method should we choose?
> How should each method be optimally tuned?

Our goal is to provide definitive answers to parts of these questions. In this paper, we take a complementary perspective to the existing literature and offer a tight, principled comparison of the three methods. Specifically, we derive their optimal tuning, identify an optimal Lyapunov function for each method, and compute the *exact* optimal convergence rate for *any* Lyapunov function within our class of candidate Lyapunov functions.

To make this comparison sharp and transparent, we adopt a deliberately simple yet representative setup: we consider smooth and strongly convex functions in the single-agent setting ($n = 1$). While simple, this regime is widely recognized as a crucial stepping stone—not only for building intuition, but also as its own theoretical contribution [e.g., 32, 10]. The single-agent setting is also of independent interest: in the field of *sparsity-aware* neural-network training, sparse-update methods have been shown to correspond to error feedback [43]. In this context, *tightness* means that we identify the best possible Lyapunov function within a given class *and* compute the exact worst-case convergence rate over the class of problems considered.

Our methodology draws on the *performance estimation* framework [44, 45], which enables the numerical derivation of exact convergence rates for a wide range of first-order methods. In particular,

recent advances [46, 47] demonstrate how to automatically search for optimal Lyapunov functions. While these approaches are primarily numerical, we build upon insights from their underlying proof structures [48] to derive new analytical results.

**Contributions.** This work makes the following contributions:

1. From a methodological perspective, this paper shows how to apply the performance estimation framework to algorithms from the *federated learning* literature that incorporate compression schemes. By leveraging this methodology—both analytically and numerically—this work paves the way for a more precise and reliable understanding of federated and distributed learning methods.

2. This work provides a *tight* analysis of EF and $EF^{21}$, and compare them with compressed gradient descent in the single-agent setting, on $L$-smooth, $\mu$-strongly convex functions. In particular, we give an analytical formula of the best possible contraction rate, by analyzing an optimal Lyapunov function within a class of candidate Lyapunov functions defined in Definition 1. Furthermore, we provide the optimal tuning for the step size in both those algorithms.

3. We demonstrate that those rates are achieved, proving that the analysis is tight.

4. We conclude that the complexities of EF and $EF^{21}$ are perfectly identical in this particular setting. Moreover, CGD outperforms both methods—both in terms of the range of settings where it converges and in terms of the optimal convergence rate achieved.

5. Finally, we contribute to the process of deriving *simple* Lyapunov functions for first-order methods, and extend known results for fixed-step methods to the setting of methods using compression.

**Paper outline.** The rest of the paper is organized as follows. In Section 2, we provide background on the relevant existing results for CGD, EF, and $EF^{21}$. We also provide the necessary background on the techniques from the performance estimation literature needed to outline the methodology we use, as well as the definition of the classes of Lyapunov functions used. Section 3 presents the main contribution of the paper: tight convergence guarantees for CGD, EF, and $EF^{21}$, along with matching lower bounds. Section 4 details the methodology we use to derive the results, and provides references to the formal results required to justify this approach. It also contains a number of numerical results that illustrate the equivalence between EF and $EF^{21}$, and performance characteristics of the three methods. Section 5 summarizes the results of the paper and provides a discussion of the results in relation to the points brought up in the introductory section.

**Notations:** We denote $\mathbb{S}^{\ell}$ the symmetric matrices, and denote $\mathbb{S}_{+}^{\ell}$ the set of positive semi definite matrices. For any two matrices $A \in \mathbb{S}^{\ell}$ and $B \in \mathbb{S}^{d}$, we denote $A \otimes B$ the Kronecker product.

## 2 Background

In this section, we briefly overview relevant existing results from the field of distributed optimization, the necessary background on the performance estimation framework, provide the rest of the assumptions we will need, and specify the notion of Lyapunov functions used in this paper.

### 2.1 Theoretical results on CGD, EF, $EF^{21}$

In the single agent case, we leverage the equivalence between compressed gradient descent (CGD), under Assumption 1 and the *inexact gradient method* with relatively bounded gradients. CGD corresponds to the particular case of Algorithm 1 with $n = 1$, and relatively bounded gradients means that for any $x$, the oracle queried at point $x$ outputs a value $g$ such that $\|g - \nabla f(x)\|^2 \leq \epsilon \|\nabla f(x)\|^2$. Various notions of gradient approximation have been studied [49–51], and a tight analysis for relatively bounded gradients was given in [52]. Specifically, authors have shown that the inexact gradient method then enjoys tight convergence guarantees for any step size $\eta > 0$, with respect to the functional residual, Euclidean norm distance to the solution and gradient norm. However, CGD is known to diverge when applied using stochastic gradient oracles, and to non-smooth functions [10]. Interestingly, it is also known to diverge in the multi-worker setting [24]. Studying CGD is important in its own right, because when the compression operator is chosen as the sign function, and the algorithm is applied in the stochastic setting (i.e., signSGD), there is a connection to Adam both in the convex [53], and non-convex setting [54].

Convergence rates for EF have been established with strongly convex [32], quasi-convex and non-convex [10, 35] functions, and using stochastic gradients. Richtárik et al. [14] study EF$^{21}$ in the multi-worker setting, with (potentially) randomized compression operators. They establish a $\mathcal{O}(k^{-1})$ convergence rate on Lipschitz smooth functions, and a linear rate under the additional assumption that the functions satisfy the Łojasiewicz inequality. These results are obtained using a Lyapunov function; however, without tightness guarantees—neither for the choice of Lyapunov function, nor for the convergence rate itself. Extensions of EF$^{21}$ have been proposed, including adaptations to stochastic gradients [23], and the introduction of a momentum term to improve sample complexity in the stochastic setting [36].

## 2.2 Performance estimation

Performance estimation tools [55, 56, 45] enable to obtain tight (i.e., exact worst-case) numerical guarantees on convergence rates for various choices of Lyapunov functions. To do so, the estimation of the worst-case rate is formulated as a semidefinite program (SDP), which is then solved numerically using standard solvers such as MOSEK [57]. The resulting numerical values approximate the exact *worst-case* rate of an algorithm over a class of functions, and should not be confused with quantities that depend on specific data, initial points, or problem instances.

This framework has been made accessible through software packages in both Python [58] and Matlab [59], enabling researchers to easily apply these tools. Advanced performance estimation techniques based on the dual formulation of the aforementioned SDP have been developed within this framework to construct optimal Lyapunov functions for first-order methods [46, 47, 60]. Particularly relevant is the approach of [46], which formulates the search for *quadratic* Lyapunov functions as a feasibility problem with a candidate contraction rate. By performing bisection on this rate, the method identifies the smallest contraction rate for which a valid Lyapunov function exists.

Another relevant line of work we leverage to discover the analytical form of the Lyapunov functions lies in the field of *symbolic regression*, which aims to solve supervised learning tasks over the space of simple analytic expressions. Recent advances use genetic programming to search this space, and software packages have been developed in both Python and Julia [61].

## 2.3 Definitions & Notation

We have already introduced the notion of a contractive compression operator in Assumption 1. To position our contribution within the broader literature we now specify that our analysis is restricted to the setting of smooth, strongly convex functions:

**Assumption 2.** *The function $f$ is $L$-smooth, i.e., for all $x, y \in \mathbb{R}^d$, we have*

$$f(y) \leqslant f(x) + \langle \nabla f(x), y - x \rangle + \frac{L}{2} \|y - x\|^2.$$

**Assumption 3.** *The function $f$ is $\mu$-strongly convex, i.e., for all $x, y \in \mathbb{R}^d$, we have*

$$f(y) \geqslant f(x) + \langle \nabla f(x), y - x \rangle + \frac{\mu}{2} \|y - x\|^2.$$

We will use the notation $\mathcal{F}_{\mu,L}$ to denote the set of smooth, strongly convex functions with parameters $\mu$ and $L$. We will denote $\kappa := \frac{L}{\mu}$ the condition number. For any objective function $f \in \mathcal{F}_{\mu,L}$, we denote $x_\star := \arg\min_{x \in \mathbb{R}^d} f$ its minimizer, and $f_\star := \min_{x \in \mathbb{R}^d} f(x)$ its minimum value.

**Lyapunov functions.** We now formally define the class of Lyapunov functions under consideration.

We formally denote $\mathcal{M} : \mathbb{R}^{\ell \times d} \times \mathbb{R}^d \times \mathcal{F} \to \mathbb{R}^{\ell \times d} \times \mathbb{R}^d$ a first-order method acting on a set of functions $\mathcal{F}$, for an integer $\ell \in \mathbb{N}$. Such a method, given a function $f \in \mathcal{F}$, is applied to an initial *state* $\xi_0 \in \mathbb{R}^{\ell \times d}$ and iterate $x_0 \in \mathbb{R}^d$, and generates a sequence $\{\xi_k\}_{k \geqslant 0}$ of states, and a sequence $\{x_k\}_{k \geqslant 0}$ of iterations. The *states* represent information summarizing the current point in the optimization trajectory that the algorithms may depend on beyond the current iterate—for example, error-related quantities in error feedback algorithms. The integer $\ell$ is thus typically small, from 0 to 3 in general. The specific states used in this paper are all specified in Subsection 3.1.

**Definition 1** (Candidate Lyapunov function). *A function $\mathcal{V} : \mathbb{R}^{\ell \times d} \times \mathbb{R}^d \to \mathbb{R}$ is called a candidate Lyapunov function for $f$ if it satisfies the following conditions:*

1. *(Non-negativity)* $\mathcal{V}(\xi, x; f) \geqslant 0$, *for any* $\xi \in \mathbb{R}^{\ell \times d}$, $x \in \mathbb{R}^d$,

2. *(Zero at fixed-point)* $\mathcal{V}(\xi, x; f) = 0$ *if and only if* $x = x_\star$ *and* $\xi = \xi_\star$ *for a unique* $\xi_\star \in \mathbb{R}^{\ell \times d}$.

3. *(Meaningfully lower bounded) there exists a positive semidefinite matrix* $A \in \mathbb{S}_+^\ell$ *and a scalar* $a \geqslant 0$ *such that* $\mathcal{V}(\xi, x; f) \geqslant (\xi - \xi_\star)^\top (A \otimes I_d)(\xi - \xi_\star) + a(f(x) - f_\star)$ *and* $\mathrm{Tr}(A) + a = 1$.

The lower bound in item 3 of our definition requires some justification: it ensures that the Lyapunov function provides control over meaningful quantities in optimization, such as the distance to the fixed point, gradient norm, algorithm-dependent quantities and the functional residual.

The class of candidate Lyapunov functions is thus given by

$$\mathbb{V}_\ell = \left\{ (P, p) \in \mathbb{S}_+^\ell \times \mathbb{R}^+ : \mathrm{Tr}(P) + p = 1 \right\}. \tag{3}$$

For any $(P, p) \in \mathbb{V}_\ell$ we denote $\mathcal{V}_{(P,p)}$ the Lyapunov functions of the form:

$$\mathcal{V}_{(P,p)}(\xi, x; f) = (\xi - \xi_\star)^\top (P \otimes I_d)(\xi - \xi_\star) + p(f(x) - f_\star). \tag{4}$$

We seek candidate Lyapunov functions $\mathcal{V} : \mathbb{R}^{\ell \times d} \times \mathbb{R}^d \times \mathcal{F} \to \mathbb{R}$ that satisfy the recurrence

$$\mathcal{V}(\xi_{k+1}, x_{k+1}; f) \leqslant \rho \cdot \mathcal{V}(\xi_k, x_k; f), \tag{5}$$

for some constant $\rho < 1$ and for all $k \geqslant 0$, uniformly over $\mathcal{F}$. Finding the *optimal* Lyapunov function within a parameterized class, for a method $\mathcal{M}$, then amounts to solving the following problem:

$$\rho^\star(\mathcal{M}) := \min_{(P,p) \in \mathbb{V}_\ell} \left\{ \max_{\substack{f \in \mathcal{F}_{\mu, L}, \\ (\xi_0, x_0) \in \mathbb{R}^{\ell \times d} \times \mathbb{R}^d}} \frac{\mathcal{V}_{(P,p)}(\xi_1, x_1; f)}{\mathcal{V}_{(P,p)}(\xi_0, x_0; f)} : (\xi_1, x_1) = \mathcal{M}(\xi_0, x_0; f) \right\}. \tag{6}$$

Note that we set $k = 0$ in order to have a guarantee that is valid for all $x \in \mathbb{R}^d$.

# 3 Main results

In this section, we provide answers to the questions stated in the introduction for our setting. We begin by showing some numerical results on the performance of each method, and then provide the precise statements of all of our theoretical results.

## 3.1 Numerical performance of all methods

In order to compare EF and EF$^{21}$ with the performance of CGD, we first need to specify the state-variables under consideration when analyzing each method. Those are given given by:

$$\xi_k^{\mathrm{CGD}} = \begin{bmatrix} x_k \\ \nabla f(x_k) \\ \mathcal{C}(\eta \nabla f(x_k)) \end{bmatrix}, \quad \xi_k^{\mathrm{EF}} = \begin{bmatrix} x_k \\ \nabla f(x_k) \\ \mathcal{C}(e_k + \eta \nabla f(x_k)) \\ e_k \end{bmatrix}, \quad \xi_k^{\mathrm{EF21}} = \begin{bmatrix} x_k \\ \nabla f(x_k) \\ d_k \end{bmatrix}, \tag{7}$$

where all variables are defined as in Algorithm 1, Algorithm 2 and Algorithm 3, respectively. Note that all numerical results of this article are using *deterministic* compression operators.

We are now ready to present the numerical results on the performance of each method. Figure 1 shows contour plots of the contraction factor for each method. That is, for a fine grid of both the step size $\eta$ in Algorithms 1 to 3, and the parameter $\epsilon$ in Assumption 1, we numerically compute the value of the best possible worst-case contraction rate in terms of our class of candidate Lyapunov functions, over $\mathcal{F}_{\mu, L}$, as given by (6). A darker blue point indicates a stronger contraction $\rho^\star(\mathcal{M})$ (i.e., a better rate). A red point indicates that the method is non-convergent for that choice of $(\epsilon, \eta)$. We observe that, although the results are purely numerical at this stage, they indicate that EF and EF$^{21}$ exhibit identical performance in our setting. This numerical equivalence is supported by Table 1: the maximum absolute difference between contraction factors for EF and EF$^{21}$ is on the order of $10^{-5}$ to $10^{-7}$. This first fact is a very surprising observation. Indeed while EF and EF$^{21}$ are known to be identical in the very specific case of using a deterministic positively homogeneous and additive

| | $\kappa = 2$ | $\kappa = 4$ | $\kappa = 10$ | | $\kappa = 2$ | $\kappa = 4$ | $\kappa = 10$ |
|---|---|---|---|---|---|---|---|
| Absolute error | 4.34e-07 | 6.02e-07 | 1.21e-06 | Absolute error | 1.14e-06 | 3.90e-07 | 5.11e-06 |

Table 1: Maximum absolute difference of contraction factor for EF and $\text{EF}^{21}$, computed over grid of $\epsilon \in [0.01, 0.99]$ and $\eta \in [0.01, \frac{2}{L+\mu}]$ for $L = 1$, and varying $\mu$.

Table 2: Maximum absolute difference of contraction factor for CGD when allowing any combination of terms in our Lyapunov function, compared to the contraction achieved by the functional residual. Same grid as Table 1.

compression operator [14, see Section 4.2], EF and $\text{EF}^{21}$ remain grounded in fundamentally different motivations at first sight: on the one hand, EF accounts for the errors introduced by the compression step, while on the other hand, $\text{EF}^{21}$ subtracts a control variate from the gradient prior to compression. Proving that the best possible convergence rate they can obtain, for any tuning $\epsilon, \eta$, is similar (but achieved for a *different* Lyapunov function) was, to the best of our knowledge, never established in the literature. It thus constitutes a significant step towards better understanding their connections.

A second observation can be made from those plots: the region of non-convergence is by far larger for EF and $\text{EF}^{21}$ than for CGD. In particular, there exist multiple tunings, for which incorporating any of the two types of error feedback, actually *prevent* convergence.

While given for a single $(\mu, L)$ in Figure 1 and Table 1, similar results hold for all values of $(\mu, L)$ that were tried numerically, and several examples are given in Appendix D, along with details of the numerical experiments, including the computation of regions of non-convergence.

Furthermore, we *tune* each algorithm by picking the optimal step size for each method. We compute the rate $\inf_\eta \rho^\star(\mathcal{M}_\eta)$, for $\mathcal{M} \in \{\text{CGD}, \text{EF}, \text{EF}^{21}\}$, where $\mathcal{M}_\eta$ corresponds to the method with step size $\eta$. Results are shown in Appendix B for three values of $\kappa$, namely 2, 4, and 10. In each setup, for every level of compression, CGD achieves a rate which is strictly better than EF and $\text{EF}^{21}$. In Appendix B.1, we also provide a symbolic certificate of this fact. These results challenge the prevailing intuition that error feedback ensures convergence comparable to that of uncompressed methods, and even demonstrate that in the single agent and deterministic gradient regime, error feedback is actually always detrimental to convergence.

Finally, we note that the functional residual constitutes an optimal Lyapunov function for CGD, as shown in Table 2. This result is not particularly surprising, given that a tight analysis of the functional residual with the optimal step size was previously established for *inexact gradient descent* by De Klerk et al. [52]. That work also demonstrated the tightness of this rate[2]. For $\text{EF}^{21}$, Richtárik

---

[2]Following the same line of reasoning as in our remark on the tightness of our Lyapunov functions in Section 5, we can then show that the functional residual is an optimal Lyapunov function for CGD.

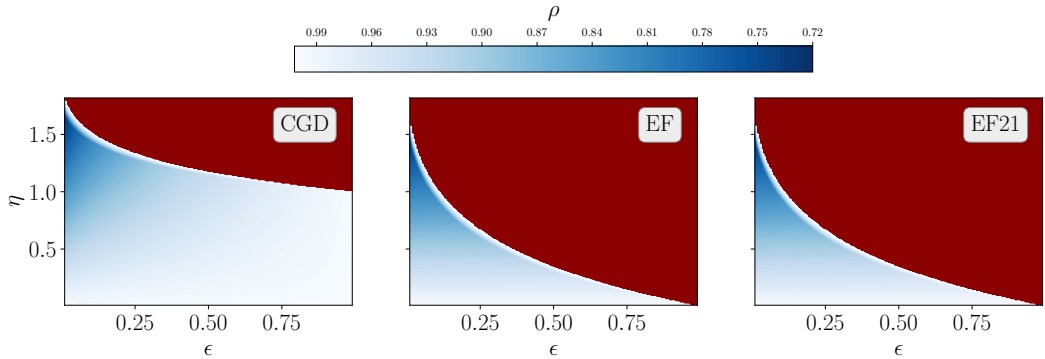

Figure 1: Single row of contour plots showing performance of CGD, EF, and $\text{EF}^{21}$ as a function of step size $\eta$ and compression parameter $\epsilon$, with regions of non-convergence marked in red. The regions of non-convergence were computed using PEPit by finding cycles of length 2.

et al. [14] proposed another Lyapunov function than the one used here. In Appendix D.5, Figure 12 we compare the complexities of an optimally tuned version of their Lyapunov function with the class-optimal Lyapunov function numerically. Further comparison of rates can be found in Appendix B. In Appendix B.2, we prove that our rate is strictly faster than the the aforementioned rate for $\text{EF}^{21}$.

In the next two sections, we provide analytical results on EF and $\text{EF}^{21}$, respectively.

### 3.2 Exact convergence rate and optimal tuning for EF

We begin by stating the main result of this section, which is a tight rate of convergence for EF.

**Theorem 1.** `CAS` `PEP`

*Consider running Algorithm 2, i.e., EF, with a compression operator $\mathcal{C}$ satisfying Assumption 1 for some $\epsilon \in [0, 1]$ on any function satisfying Assumptions 2, and 3. Let the step size be given by*

$$\eta^\star = \left(\frac{2}{L+\mu}\right) \cdot \left(\frac{1-\sqrt{\epsilon}}{1+\sqrt{\epsilon}}\right). \tag{8}$$

*Then, we have that*

$$\rho^\star(\text{EF}_{\eta^\star}) = \sqrt{\epsilon} + \tfrac{1}{4}(1+\sqrt{\epsilon})(L-\mu)\lambda, \tag{9}$$

*where*

$$\lambda := \tfrac{\eta^\star}{L+\mu}\left[(1-\sqrt{\epsilon})(L-\mu) + (1+\sqrt{\epsilon})\sqrt{(L-\mu)^2 + 16L\mu\frac{\sqrt{\epsilon}}{(1+\sqrt{\epsilon})^2}}\right]. \tag{10}$$

*A Lyapunov function achieving the rate in (9), with $\xi^{\text{EF}}$ defined in (7), is given by*

$$\mathcal{V}(\xi^{\text{EF}}, x; f) := \|x - x_\star\|^2 - 2(x-x_\star)^\top e + \left(1 + \frac{1}{\sqrt{\epsilon}}\right) \cdot \|e\|^2 = \|x - x_\star - e\|^2 + \frac{1}{\sqrt{\epsilon}}\|e\|^2, \tag{11}$$

*Finally, the step size in (8) is worst-case optimal for EF: $\forall \eta \geqslant 0$, we have $\rho^\star(\text{EF}_\eta) \geqslant \rho^\star(\text{EF}_{\eta^*})$.*

Importantly, (9) shows that the rate is tight; that is, there exist $f \in \mathcal{F}_{\mu,L}$ and $(P, p) \in \mathbb{V}_\ell$ for which the rate is exactly achieved. Since the lower bound also applies to any other performance measure in our state space, our performance measure is optimal. This is formally demonstrated in the proof provided in Appendix C.1. The proof was written using *deterministic* compressors, but the same guarantees hold under expectation with stochastic compressors as is explained in Appendix C.3.

For completeness—and to support both our theoretical results and the tightness of the numerical results obtained through performance estimation—we provide additional figures comparing the empirically observed optimal step sizes for worst-case instances with our theoretical step size $\eta^\star$, across different values of $\epsilon$ and $\mu/L$. These results are shown in Figure 11, located in Appendix D.4. The numerical and analytical values match up to numerical accuracy.

### 3.3 Exact convergence rate and optimal tuning for $\text{EF}^{21}$

We now state our main result on the $\text{EF}^{21}$ algorithm, which is also tight.

**Theorem 2.** `CAS` `PEP`

*Consider running Algorithm 3 with a compression operator satisfying Assumption 1 for some $\epsilon \in [0, 1]$ on any function satisfying Assumptions 2, and 3. Let the step size be given by $\eta^\star$ in (8). Then,*

$$\rho^\star(\text{EF}^{21}{}_{\eta_\star}) = \rho^\star(\text{EF}_{\eta_\star}). \tag{12}$$

*A Lyapunov function achieving the rate in (12) is given by*

$$\mathcal{V}(\xi^{\text{EF}^{21}}, x; f) := (1+\sqrt{\epsilon}) \cdot \|g\|^2 - 2g^\top d + \|d\|^2 = \|g - d\|^2 + \sqrt{\epsilon} \cdot \|g\|^2. \tag{13}$$

*Finally, the step size $\eta^\star$ is worst-case optimal for this algorithm.*

**Remark 1.** *The explicit rate for EF and $\text{EF}^{21}$ can be written as*

$$\rho = \sqrt{\epsilon} + \left(\frac{1-\sqrt{\epsilon}}{2}\right)\left(\frac{\kappa-1}{\kappa+1}\right)^2 \left[1 - \sqrt{\epsilon} + \sqrt{(1+\sqrt{\epsilon})^2 + \sqrt{\epsilon}16\frac{\kappa}{(\kappa-1)^2}}\right], \tag{14}$$

*where $\kappa = L/\mu$. A detailed comparison between this and the existing rate for $\text{EF}^{21}$ under the Łojasiewicz inequality [14] is provided in Appendix B.*

The proof of this theorem is given in Appendix C.2 and assumes *deterministic* compressors, but the same guarantees hold under expectation with stochastic compressors, as is explained in Appendix C.3.

This second theoretical result analytically confirms the surprising numerical observation from Subsection 3.1, illustrated in Figure 1: EF and $\text{EF}^{21}$ have the exact same optimal guarantee. Furthermore, the optimal step size is also the same, and can also be argued to be worst-case optimal in our setting using the same arguments as in Section 3.2. However, the optimal Lyapunov function is very different.

### 3.4 Tightness over multiple iterations and choice of state variables

A natural question arising from the above analysis concerns the tightness of the Lyapunov functions provided in Theorems 1 and 2 *over multiple steps*. Specifically, for $K \geqslant 2$, we investigate whether the convergence rate of a method $\mathcal{M} \in \text{EF}, \text{EF}^{21}, \text{CGD}$ can be improved by analyzing $\mathcal{M}^K$, the method run over $K$ iterations, defined as follows:

$$\rho_K^\star(\mathcal{M}) := \left( \min_{(P,p)\in\mathbb{V}_\ell} \left\{ \max_{\substack{f\in\mathcal{F}_{\mu,L}, \\ (\xi_0,x_0)\in\mathbb{R}^{\ell\times d}\times\mathbb{R}^d}} \frac{\mathcal{V}_{(P,p)}(\xi_K, x_K; f)}{\mathcal{V}_{(P,p)}(\xi_0, x_0; f)} : (\xi_K, x_K) = \mathcal{M}^K(\xi_0, x_0; f) \right\} \right)^{1/K}.$$

We provide a numerical answer to that question in Appendix D.2 for all algorithms discussed in the paper, showing that the analysis of the single-state Lyapunov functions used in this work are tight even if we consider multiple iterations. To that end, we plot the worst-case contractions for multiple iterations computed using PEPit [58], on the optimal Lyapunov functions given by (11) and (13).

**Lyapunov Tightness.** To prove that the Lyapunov functions we use in Theorems 1 and 2 are tight, one has to show that the rate of convergence for any other candidate Lyapunov function from our class is lower bounded by the rate we obtain. We consider a quadratic function, used in the latter sections of our proofs to prove tightness: asymptotic expressions for all the state-variables are the same, up to an iteration-independent constant. Consequently, when computing ratios between any set of Lyapunov functions, these constants cancel out. The only thing remaining is the term that actually depends the iteration count, which is exactly equal to the rate given in Theorems 1 and 2.

## 4 Methodology

We now present the methodology used to obtain the results in Section 3. The approach builds on the framework developed by [46]. We extend it to cover methods using *deterministic* compression operators under Assumption 1 in Appendix A. Obtaining the proofs of Theorems 1 and 2 required a combination of advanced performance estimation techniques (finding optimal Lyapunov functions), several tricks, as well as symbolic computation and symbolic regression frameworks.

To solve problem (6), we begin by addressing the inner maximization problem for a fixed contraction factor $\rho$. This amounts to checking the feasibility of a semidefinite program, detailed in Appendix A. We then apply bisection on $\rho$ to identify the smallest admissible contraction factor. However, the Lyapunov function given is rarely unique, and most solutions obtained numerically vary significantly with problem parameters such as the compression factor $\epsilon$. To address this, we use rank minimization heuristics—specifically, the logdet heuristic [62]. This enables one to obtain a unique set of structurally simpler, low-rank Lyapunov functions. Finally, we proceed by eliminating redundant coefficients in the matrix $P$ and the scalar $p$, to arrive at the concise forms presented in Section 3. At the end of this process, the Lyapunov function coefficients were found to be mutually dependent, reducing the problem to identifying a closed-form expression for any one of them. To estimate such a coefficient, we applied symbolic regression using the PySR Python package [61]. This approach proved highly effective at finding simple yet *optimal* Lyapunov functions. To arrive at simple and readable proofs, we leverage the computer algebra system of *Mathematica* [63].

We wish to emphasize that the combined use of log-det heuristics, symbolic regression, and a computer algebra system turned out to be highly effective at solving this problem, and we believe it has broad applicability to other problems in machine learning.

# 5   Conclusion

In this paper, we provided tight analyses of EF and $\text{EF}^{21}$ using Lyapunov functions, with guarantees on both the Lyapunov functions themselves and the convergence rates achieved. Notably, both algorithms exhibit the same convergence rate in our setting, and through a remark made in the discussion below, this gives us tight rates for any of the candidate Lyapunov functions we considered in our class. We also observed that their performance is strictly worse than that of compressed gradient descent—an outcome that, we believe, challenges the intuition of many in the field.

Our analysis is confined to the single-agent setting, both as an interesting problem on its own, and as a source of intuition for the multi-agent case. CGD cannot serve as a baseline in the multi-agent setting as it fails to converge with more than one agent [24]. In contrast, $\text{EF}^{21}$ was specifically designed to *improve* convergence over EF in the multi-agent setting. Yet, its convergence rate in the single-agent case matches exactly that of EF. The findings of this paper raise two compelling questions:

- *Does the performance of* EF *and* $\text{EF}^{21}$ *differ in the multi-agent setting?*
- *Are there more effective error compensation mechanisms yet to be discovered?*

We leave these questions for future work and conclude by emphasizing that the methodology used is likely applicable to a broad range of problems in optimization for machine learning. We look forward to seeing it extended and applied in future research.

## Acknowledgments and Disclosure of Funding

We would like to thank Si Yi Meng, Jean-Baptiste Fest, Abel Douzal, and Lucas Versini for providing helpful feedback on an early draft of this paper. D. Berg Thomsen and A. Taylor are supported by the European Union (ERC grant CASPER 101162889). The work of A. Dieuleveut is partly supported by ANR-19-CHIA-0002-01/chaire SCAI, and Hi!Paris FLAG project, PEPR Redeem. The French government also partly funded this work under the management of Agence Nationale de la Recherche as part of the "France 2030" program, references ANR-23-IACL-0008 "PR[AI]RIE-PSAI", ANR-23-PEIA-005 (REDEEM project) and ANR-23-IACL-0005.

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

## Organization of the appendix

This appendix provides additional content and details complementing the paper. In particular, Appendix A details the general methodology used to search for Lyapunov functions. The complete missing proofs for the main results of the paper are presented in Appendix C. Appendix D presents additional numerical results that informally motivate a few choices made in the paper and provide numerical validation of our claims. Finally, Appendix E discusses our choice of annotating this article with proof certificates.

## A  Feasibility problems with compressors

This section presents the methodology used to search for Lyapunov functions. In a nutshell, we formulate the Lyapunov search problem as a quasi-convex optimization problem involving linear matrix inequalities. Those problems are typically solved through the use of an iterative procedure involving a binary search with semidefinite solvers—we use MOSEK [57] throughout. The main steps taken here can be viewed as a generalization of the procedure proposed in [46] to first-order methods using compression, in particular Algorithms 2 and 3. We also simplify a few steps that are not needed for our purposes. We start by reviewing the technique on a simpler example in Appendix A.1 before detailing the more tricky formulations involving compression.

### A.1  Feasibility problem for gradient descent

To introduce the concepts underlying the techniques used to construct Lyapunov functions for Algorithms 2 and 3, we begin with the simpler case of *gradient descent* on smooth, strongly convex functions. That is, we consider the algorithm:

$$x_1 = x_0 - \eta \nabla f(x_0). \tag{GD$_\eta$}$$

The goal of this subsection is to review the steps used to compute $\rho^\star(\mathrm{GN}_\eta)$, as defined in (6), via a bisection search in which each iteration involves verifying the feasibility of a convex problem.

Our starting point is to consider the following state variable for GD:

$$\xi_k^{\mathrm{GD}} = \begin{bmatrix} x_k \\ \nabla f(x_k) \end{bmatrix} \tag{15}$$

and a natural family of Lyapunov function candidates (which corresponds to a subset of (3)) of the form

$$\begin{aligned}
\mathcal{V}_P(\xi_k^{\mathrm{GD}}, x_k; f) \equiv \mathcal{V}_P(x_k, \nabla f(x_k); f) &:= \begin{bmatrix} x_k - x_\star \\ \nabla f(x_k) \end{bmatrix}^\top (P \otimes I_d) \begin{bmatrix} x_k - x_\star \\ \nabla f(x_k) \end{bmatrix} \\
&= P_{11} \|x_k - x_\star\|^2 + P_{22} \|\nabla f(x_k)\|^2 \\
&\quad + 2P_{12} \langle \nabla f(x_k); x_k - x_\star \rangle,
\end{aligned} \tag{16}$$

where $P \in \mathbb{S}^d$ is positive semidefinite and we require $\mathrm{Tr}(P) = 1$. This latter requirement is without loss of generality due to a normalization argument, and is added to avoid the trivial solution $P = 0$.

The problem we aim to solve is that of finding the best Lyapunov function among a given set of candidates—specifically, the one for which the ratio $\frac{\mathcal{V}_P(x_1, \nabla f(x_1))}{\mathcal{V}_P(x_0, \nabla f(x_0))}$ can be uniformly upper bounded by the smallest possible constant over all optimization problems in the considered family. In other words, we seek the Lyapunov candidate function that yields the smallest possible $\rho$ such that

$$\frac{\mathcal{V}_P(x_1, \nabla f(x_1); f)}{\mathcal{V}_P(x_0, \nabla f(x_0); f)} \leqslant \rho \tag{17}$$

is valid for all $L$-smooth $\mu$-strongly convex functions $f : \mathbb{R}^d \to \mathbb{R}$ (in any dimension $d \in \mathbb{N}$) and all possible $x_0, x_1, x_\star \in \mathbb{R}^d$ compatible with $f$ and $x_1 = x_0 - \eta \nabla f(x_0)$. The problem can be phrased as finding

$$\rho^\star(\mathrm{GD}_\eta) = \min_{P \succcurlyeq 0} \left( \max_{\substack{d \in \mathbb{N} \\ f \in \mathcal{F}_{\mu,L} \\ x_0, x_\star \in \mathbb{R}^d}} \left\{ \frac{\mathcal{V}_P(x_1, \nabla f(x_1); f)}{\mathcal{V}_P(x_0, \nabla f(x_0); f)} \quad \text{s.t.} \quad \mathrm{Tr}(P) = 1, \ x_1 = x_0 - \eta \nabla f(x_0) \right\} \right) \tag{18}$$

where $\eta > 0$ is the step size. One can reformulate this problem directly using tools from the performance estimation literature [55, 56], but the resulting problem is not convex in the variable $\rho$. One way to address this is to reduce it to the problem of finding, for a given contraction factor $\rho$, some Lyapunov function that achieves this $\rho$—if one exists. This problem, on the other hand, is convex, and we can simply perform bisection search on $\rho$ to find the smallest possible contraction factor.

We now introduce some notation to simplify the statement of finding such Lyapunov functions:

$$\sigma_\rho(x_1, g_1, x_0, g_0; P) := \mathcal{V}_P(x_1, g_1; f) - \rho \mathcal{V}_P(x_0, g_0; f). \tag{19}$$

We will arrive at a way of solving this problem by reasoning through two steps:

1. **Step 1:** verifying a *given* Lyapunov function and rate as a convex problem.

2. **Step 2:** verifying a rate $\rho$ for the *optimal* candidate Lyapunov functions as a convex problem.

**Step 1: verifying a *given* Lyapunov function and rate as a convex problem.**
For a fixed Lyapunov parameter $P \in \mathbb{V}_2$ and a tentative rate $\rho > 0$, we can then state the problem of *verifying* a given Lyapunov function as that of showing that the minimum value of the following is non-positive:

$$0 \geqslant \sup_{\substack{d \in \mathbb{N} \\ f \in \mathcal{F}_{\mu,L} \\ x_0, x_\star \in \mathbb{R}^d}} \sigma_\rho(x_1, \nabla f(x_1), x_0, \nabla f(x_0); P; f)$$

$$\text{s.t.} \quad x_1 = x_0 - \eta \nabla f(x_0) \tag{20}$$
$$\nabla f(x_\star) = 0$$

The constraint that $f$ is a smooth strongly convex function is easily encoded using interpolation conditions [56]. This allows us to work with *sampled points* from $f$ rather than the infinite dimensional set $\mathcal{F}_{\mu,L}$. We introduce the notation

$$\phi_{ij} := f_i - f_j - g_j^\top (x_i - x_j) - \frac{1}{2L} \|g_i - g_j\|^2 - \frac{\mu}{2(1 - \mu/L)} \left\| x_i - x_j - \frac{1}{L}(g_i - g_j) \right\|^2, \quad (21)$$

where the notation $(x_i, g_i, f_i)$ is used to denote a sampled triplet from $f$, such that $f(x_i) = f_i$ and $\nabla f(x_i) = g_i$ for all $i \in \{0, 1, \star\}$. This lets us rephrase our problem as

$$0 \geqslant \sup_{\substack{d \in \mathbb{N} \\ x_\star, x_0, g_\star, g_0, g_1 \in \mathbb{R}^d \\ f_0, f_1 \in \mathbb{R}}} \sigma_\rho(x_1, g_1, x_0, g_0; P; f)$$

$$\text{s.t.} \quad \phi_{ij} \geqslant 0 \qquad \forall i, j \in \{0, 1, \star\} \tag{22}$$
$$x_1 = x_0 - \eta \nabla f(x_0)$$
$$g_\star = 0$$

The above problem is not convex due to the interpolation constraints. To address this, we reformulate it as a semidefinite program (SDP). Let $G = B^\top B$ where $B$ is the following $(3 \times d)$ matrix

$$B = [x_0 - x_\star, \ g_0, \ g_1],$$

and let $\mathbf{f} := \begin{bmatrix} f_0 - f_\star \\ f_1 - f_\star \end{bmatrix}$. In other words, $G \succcurlyeq 0$ is the Gram matrix of the entries of $B$:

$$\begin{bmatrix} \|x_0 - x_\star\|^2 & \langle g_0, x_0 - x_\star \rangle & \langle g_1, x_0 - x_\star \rangle \\ \langle g_0, x_0 - x_\star \rangle & \|g_0\|^2 & \langle g_1, g_0 \rangle \\ \langle g_1, x_0 - x_\star \rangle & \langle g_1, g_0 \rangle & \|g_1\|^2 \end{bmatrix} \succcurlyeq 0.$$

We introduce a convenient notation by defining basis (row) vectors in $\bar{x}_i, \bar{g}_i \in \mathbb{R}^3$ and $\bar{f}_i \in \mathbb{R}^2$ which allow us to "select" specific elements in $B$ and $\mathbf{f}$. Specifically, we define them such that

$$x_i - x_\star = B\bar{x}_i^\top, \qquad g_i = B\bar{g}_i^\top, \qquad f_i - f_\star = \bar{f}_i\mathbf{f}. \tag{23}$$

More precisely: $\bar{x}_i = \mathbf{e}_1^\top \in \mathbb{R}^3$, $\bar{g}_0 = \mathbf{e}_2^\top \in \mathbb{R}^3$, $\bar{g}_1 = \mathbf{e}_3^\top \in \mathbb{R}^3$, $\bar{x}_1 = \bar{x}_0 - \eta\bar{g}_0$, $\bar{x}_\star = \mathbf{0}_3^\top \in \mathbb{R}^3$ along with $\bar{f}_0 = \mathbf{e}_1^\top \in \mathbb{R}^2$, $\bar{f}_1 = \mathbf{e}_2^\top \in \mathbb{R}^2$, $\bar{f}_\star = \mathbf{0}_2^\top \in \mathbb{R}^2$. We can conveniently rewrite the different parts of problem (22) using the notation

$$\|x_i - x_\star\|^2 = \bar{x}_i B^\top B \bar{x}_i^\top = \mathrm{Tr}(\bar{x}_i^\top \bar{x}_i G),$$
$$\|g_i\|^2 = \bar{g}_i B^\top B \bar{g}_i^\top = \mathrm{Tr}(\bar{g}_i^\top \bar{g}_i G),$$
$$\langle g_i, x_j - x_\star \rangle = \bar{g}_i B^\top B \bar{x}_j^\top = \mathrm{Tr}\left((\bar{g}_i \odot \bar{x}_j)G\right)$$

where $\bar{g}_i \odot \bar{x}_j = \frac{1}{2}(\bar{g}_i^\top \bar{x}_j + \bar{x}_j^\top \bar{g}_i)$, and $\odot$ denotes the symmetric outer product. This notation allows us to express all relevant quantities in terms of traces involving symmetric matrices. We now reformulate the necessary terms to derive the desired semidefinite representation of the problem. Let us begin with

$$\mathcal{V}_P(x_0, \nabla f(x_0); f) = P_{11}\,\mathrm{Tr}(\bar{x}_0^\top \bar{x}_0 G) + P_{22}\,\mathrm{Tr}(\bar{g}_0^\top \bar{g}_0 G) + 2P_{12}\,\mathrm{Tr}((\bar{x}_0 \odot \bar{g}_0)G)$$
$$= \mathrm{Tr}(A_0^\top P A_0 G),$$

where $A_0 := \begin{bmatrix} \bar{x}_0 \\ \bar{g}_0 \end{bmatrix}$. Similarly, we can write $\mathcal{V}_P(x_1, \nabla f(x_1); f) = \mathrm{Tr}(A_1^\top P A_1 G)$ with $A_1 := \begin{bmatrix} \bar{x}_1 \\ \bar{g}_1 \end{bmatrix}$ and also define the matrices $M_{ij}$ for all $i,j \in \{0,1,\star\}$ such that $\phi_{ij} = m_{ij}\mathbf{f} + \mathrm{Tr}(M_{ij}G)$:

$$m_{ij} = \bar{f}_i - \bar{f}_j$$
$$M_{ij} = -\bar{g}_j \odot (\bar{x}_i - \bar{x}_j) - \frac{1}{2L}(\bar{g}_i - \bar{g}_j)^\top(\bar{g}_i - \bar{g}_j) \tag{24}$$
$$\qquad - \frac{\mu}{2(1-\mu/L)}(\bar{x}_i - \bar{x}_j - \frac{1}{L}(\bar{g}_i - \bar{g}_j))^\top(\bar{x}_i - \bar{x}_j - \frac{1}{L}(\bar{g}_i - \bar{g}_j))$$

This enables a convenient reformulation of (22) as the verification of the following condition, which corresponds to solving a standard semidefinite program:

$$0 \geqslant \sup_{\substack{G \succcurlyeq 0 \\ \mathbf{f}}} \quad \mathrm{Tr}(A_1^\top P A_1 G) - \rho\,\mathrm{Tr}(A_0^\top P A_0 G)$$
$$\text{s.t.} \quad \mathrm{Tr}(M_{ij}G) + \mathbf{f}^\top c_{ij} \geqslant 0 \qquad \forall i,j \in \{0,1,\star\}. \tag{25}$$

One immediate consequence is that the validity of a given Lyapunov function $\mathcal{V}_P$ for a specified rate $\rho$—satisfying (17) or, equivalently, (20)—can be formulated as the convex problem (25).

**Step 2: verifying a rate $\rho$ for the *optimal* candidate Lyapunov functions as a convex problem.**
To derive a convenient condition that formally guarantees the above problem is nonpositive, we consider its standard Lagrangian dual[3]—which reduces to verifying the existence of dual variables $\lambda_{ij} \geqslant 0$ such that

$$0 \geqslant \inf_{\lambda_{ij} \geqslant 0} \quad 0$$
$$\text{s.t.} \quad A_1^\top P A_1 - \rho A_0^\top P A_0 - \sum_{i,j \in \{0,1,\star\}} \lambda_{ij} M_{ij} \succcurlyeq 0 \tag{26}$$
$$\sum_{i,j \in \{0,1,\star\}} \lambda_{ij} m_{ij} = 0.$$

The problem has finally reduced to showing that for a given matrix $P$, the small-sized problem (26) is feasible.

**Key idea:** As this problem is linear in $P$ (when $\rho$ is fixed), we can directly use it to search for a valid $P$ that verifies the decrease condition in (5) for a given $\rho$:

$$\underset{\substack{P \succcurlyeq 0, \\ \lambda_{ij} \geqslant 0}}{\text{feasible}} \begin{cases} A_1^\top P A_1 - \rho A_0^\top P A_0 - \sum_{i,j} \lambda_{ij} M_{ij} \succcurlyeq 0 \\ \sum_{i,j} \lambda_{ij} m_{ij} = 0 \\ \mathrm{Tr}(P) = 1. \end{cases} \tag{GD-SDP}$$

[3]Strong duality holds in this case due to the existence of a Slater point; see, e.g., [56].

**Conclusion.** Problem (GD-SDP) is a convex feasibility problem that encodes the existence of a candidate Lyapunov function certifying a given rate $\rho$. By performing a bisection search over $\rho$, we can identify the smallest rate satisfied by *some* Lyapunov function in our class. This approach enables us to solve the problem numerically using a semidefinite solver.

## A.2 Feasibility problem for EF

Using the same ideas as in Appendix A.1, this section states the feasibility problem we solve to identify Lyapunov functions for Algorithm 2. In short, this requires adapting the two steps presented in Appendix A.1 to accommodate the compressed message. Practically speaking, **Step 1** must be adapted to a slightly larger problem (with a few more states both in the Gram matrix $G$ and in the Lyapunov candidates to incorporate compression). Then, **Step 2** follows directly by the same reasoning as before: the condition derived in Step 1 reduces to checking the feasibility of a linear matrix inequality that is linear in $(P, p)$, which in turn allows us to search for the Lyapunov function via binary search over $\rho$.

Recall that we defined our state space for EF in (7) as

$$
\xi_i^{\mathrm{EF}} = \begin{bmatrix} x_i \\ \nabla f(x_i) \\ \mathcal{C}(e_i + \eta \nabla f(x_i)) \\ e_i \end{bmatrix}.
$$

This space has dimension 4, and our normalized set of candidate Lyapunov functions is given by

$$
\mathbb{V}_4 = \left\{ (P, p) \in \mathbb{S}_+^4 \times \mathbb{R}^+ : \mathrm{Tr}(P) + p = 1 \right\}.
$$

For any $(P, p) \in \mathbb{V}_4$ we thus consider $\mathcal{V}_{(P,p)}$ Lyapunov functions of the form:

$$
\mathcal{V}_{(P,p)}(\xi^{\mathrm{EF}}, x; f) = (\xi^{\mathrm{EF}} - \xi_\star^{\mathrm{EF}})^\top (P \otimes I_d)(\xi^{\mathrm{EF}} - \xi_\star^{\mathrm{EF}}) + p(f(x) - f_\star). \tag{27}
$$

where we impose $\mathrm{Tr}(P) + p = 1$, again, without loss of generality and to avoid the trivial solution $(P, p) = 0$. Similarly to (19), we now define

$$
\sigma_\rho^{\mathrm{EF}}(\xi_1^{\mathrm{EF}}, \xi_0^{\mathrm{EF}}; (P,p); f) := \mathcal{V}_{(P,p)}(\xi_1^{\mathrm{EF}}, x_1; f) - \rho \mathcal{V}_{(P,p)}(\xi_0^{\mathrm{EF}}, x_0; f). \tag{28}
$$

Again, we say that for $(P, p) \in \mathbb{V}_4$, a Lyapunov function $\mathcal{V}_{(P,p)}$ *satisfies rate $\rho$* for the iterates of EF if we have that

$$
\begin{aligned}
0 \geqslant \sup_{\substack{d \in \mathbb{N} \\ f \in \mathcal{F}_{\mu,L} \\ x_0, x_\star \in \mathbb{R}^d}} \quad & \sigma_\rho^{\mathrm{EF}}(\xi_1^{\mathrm{EF}}, \xi_0^{\mathrm{EF}}; (P,p); f) \\
\text{s.t.} \quad & (x_1, \xi_1^{\mathrm{EF}}) = \mathrm{EF}(x_0, \xi_0^{\mathrm{EF}}; f) \\
& \nabla f(x_\star) = 0
\end{aligned}
\tag{29}
$$

Formally, we require the following lemma:

**Lemma 1** (EF feasibility problem). *Consider running Algorithm 2 with a deterministic compression operator satisfying Assumption 1 for some $\epsilon \in [0, 1]$ on any function satisfying Assumptions 2 and 3. There exists a nonzero candidate Lyapunov function $\mathcal{V}_{(P,p)}$ of the form defined in (27), satisfying a given rate $\rho > 0$, if and only if the following problem is feasible:*

$$
\underset{\substack{P \in \mathbb{S}^4, \\ p \in \mathbb{R}, \\ \lambda_{ij} \geqslant 0, \\ \nu_i \geqslant 0}}{\textit{feasible}} \left\{
\begin{aligned}
& 0 \succcurlyeq \Delta V_P(\rho) + \sum_{i,j \in \{0,1,\star\}} \lambda_{ij} M_{ij} + \sum_{i \in \{0,1\}} \nu_i \cdot C_i^{\mathrm{EF}} \\
& 0 \geqslant_2 \Delta v_p(\rho) + \sum_{i,j \in \{0,1,\star\}} \lambda_{ij} m_{ij} \\
& 0 \preccurlyeq P \\
& 0 \leqslant p \\
& 1 = \mathrm{Tr}(P) + p
\end{aligned}
\right.
\tag{EF-SDP}
$$

*where matrices $(M_{ij})_{i,j \in \{0,1,\star\}}$ are defined as in (24), $(C_i^{\mathrm{EF}})_{i \in \{0,1\}}$ given below in (36), and $\Delta V_P(\rho), \Delta v_p(\rho)$ are given below in (33) and (34). Here $\geqslant_2$ denotes coordinate-wise inequality in $\mathbb{R}^2$.*

*Proof sketch.* The proof is decomposed into several steps that correspond to adapting the technical ingredients from Appendix A.1.

**Basis vector encoding** We begin by introducing notation analogously to (23) for EF. Define the following basis vectors $\bar{x}_i, \bar{g}_i, \bar{c}_i, \bar{e}_i \in \mathbb{R}^8$:

$$\bar{x}_i := \mathbf{e}_{i+1}^\top, \quad \bar{g}_i := \mathbf{e}_{i+3}^\top, \quad \bar{c}_i := \mathbf{e}_{i+5}^\top, \quad \bar{e}_i := \mathbf{e}_{i+7}^\top, \quad i \in \{0, 1\}, \tag{30}$$

where $\mathbf{e}_i$ is the $i$-th basis vector in dimension 8. Similarly, let $\bar{f}_i \in \mathbb{R}^2$ be defined by

$$\bar{f}_i := \mathbf{e}_i^\top, \quad i \in \{0, 1\}, \tag{31}$$

where $\mathbf{e}_i$ is the $i$-th basis vector in dimension 2.

The point of defining these vectors is the same as it was for GD. These vectors allow us to "select" points from our Gram matrix (see (23)). This, in turn, allows us to express the interpolation conditions of our feasibility problem in a clean manner.

We also define row vectors that correspond to the fixed-point as:

$$\bar{x}_\star := \mathbf{0}_8^\top, \quad \bar{g}_\star := \mathbf{0}_8^\top, \quad \bar{c}_\star := \mathbf{0}_8^\top, \quad \bar{e}_\star := \mathbf{0}_8^\top,$$
$$\bar{f}_\star := \mathbf{0}_2^\top.$$

Finally, we define our method in terms of our basis vectors:

$$\begin{aligned} \bar{x}_1 &= \bar{x}_0 - \bar{c}_0, \\ \bar{e}_1 &= \bar{e}_0 + \eta \bar{g}_0 - \bar{c}_0. \end{aligned} \tag{32}$$

**Expressing (28) using basis vectors.** First, we encode the decrease in the linear and quadratic terms of (28) as

$$\Delta V_P(\rho) := \begin{bmatrix} \bar{x}_1 - \bar{x}_\star \\ \bar{g}_1 - \bar{g}_\star \\ \bar{c}_1 - \bar{c}_\star \\ \bar{e}_1 - \bar{e}_\star \end{bmatrix}^\top P \begin{bmatrix} \bar{x}_1 - \bar{x}_\star \\ \bar{g}_1 - \bar{g}_\star \\ \bar{c}_1 - \bar{c}_\star \\ \bar{e}_1 - \bar{e}_\star \end{bmatrix} - \rho \begin{bmatrix} \bar{x}_0 - \bar{x}_\star \\ \bar{g}_0 - \bar{g}_\star \\ \bar{c}_0 - \bar{c}_\star \\ \bar{e}_0 - \bar{e}_\star \end{bmatrix}^\top P \begin{bmatrix} \bar{x}_0 - \bar{x}_\star \\ \bar{g}_0 - \bar{g}_\star \\ \bar{c}_0 - \bar{c}_\star \\ \bar{e}_0 - \bar{e}_\star \end{bmatrix}, \tag{33}$$

$$\Delta v_p(\rho) := p \left( \bar{f}_1 - \bar{f}_\star \right) - \rho \cdot p \left( \bar{f}_0 - \bar{f}_\star \right), \tag{34}$$

where $\rho > 0$ is the contraction factor to be verified. Note that $\Delta V_P(\rho) \in \mathbb{R}^{8 \times 8}$, and $\Delta v_p(\rho) \in \mathbb{R}^2$.

Using these objects, we have that:

$$\sigma_\rho^{\mathrm{EF}}(\xi_1^{\mathrm{EF}}, \xi_0^{\mathrm{EF}}; (P, p); f) = \mathrm{Tr} \left( (\Delta V_P(\rho)) G^{\mathrm{EF}} \right) + (\Delta v_p(\rho))^\top F^{\mathrm{EF}}, \tag{35}$$

where $G^{\mathrm{EF}} = (B^{\mathrm{EF}})^\top B^{\mathrm{EF}}$ is the Gram matrix of vectors

$$B^{\mathrm{EF}} = [x_0, x_1, \nabla f(x_0), \nabla f(x_1), \mathcal{C}(e_0 + \eta \nabla f(x_0)), \mathcal{C}(e_1 + \eta \nabla f(x_1)), e_0, e_1],$$

and $F^{\mathrm{EF}} = (f(x_0), f(x_1))$.

**Interpolation conditions** The interpolation conditions that enforce $f \in \mathcal{F}_{\mu, L}$ are identical to those we define in (24), but using the new basis vectors we define specifically for EF. We do, however, need to introduce a new interpolation condition to encode the fact that we are using a contractive compressor. We do this for *deterministic* compression operators. Using our basis vectors, this corresponds to introducing the matrices

$$C_i^{\mathrm{EF}} = (\eta \bar{g}_i + \bar{e}_i - \bar{c}_i)^\top (\eta \bar{g}_i + \bar{e}_i - \bar{c}_i) - \epsilon \cdot (\eta \bar{g}_i + \bar{e}_i)^\top (\eta \bar{g}_i + \bar{e}_i), \tag{36}$$

for $i \in \{0, 1\}$.

Finally, following the same reasoning as described in **Step 1** and **Step 2** of Appendix A.1, and using the technical modifications we outlined here, we arrive at the feasibility problem described in the statement of the lemma. $\qquad \square$

### A.3  Feasibility problem for $\mathrm{EF}^{21}$

As in the previous section, one can now adapt the same ideas as in Appendix A.1 and Appendix A.2 to $\mathrm{EF}^{21}$. This section states the feasibility problem we solve to identify Lyapunov functions in this context.

Following a parallel line of derivation, we begin by noting that the state space defined in (7) for $\mathrm{EF}^{21}$ was

$$\xi_k^{\mathrm{EF}^{21}} = \begin{bmatrix} x_k \\ \nabla f(x_k) \\ d_k \end{bmatrix}, \tag{37}$$

This space now has dimension 3, and our normalized set of candidate Lyapunov functions is given by

$$\mathbb{V}_3 = \left\{ (P, p) \in \mathbb{S}_+^3 \times \mathbb{R}^+ : \mathrm{Tr}(P) + p = 1 \right\}.$$

For any $(P, p) \in \mathbb{V}_3$ we thus consider $\mathcal{V}_{(P,p)}$ Lyapunov functions of the form:

$$\mathcal{V}_{(P,p)}(\xi^{\mathrm{EF}^{21}}, x; f) = (\xi^{\mathrm{EF}^{21}} - \xi_\star^{\mathrm{EF}^{21}})^\top (P \otimes I_d)(\xi^{\mathrm{EF}^{21}} - \xi_\star^{\mathrm{EF}^{21}}) + p(f(x) - f_\star). \tag{38}$$

where we once more require $\mathrm{Tr}(P) + p = 1$ without loss of generality and to avoid the trivial solution $(P, p) = 0$. Similarly, to (19) and (28), we define here

$$\sigma_\rho^{\mathrm{EF}^{21}}(\xi_1^{\mathrm{EF}^{21}}, \xi_0^{\mathrm{EF}^{21}}; (P, p); f) := \mathcal{V}_{(P,p)}(\xi_1^{\mathrm{EF}^{21}}, x_1; f) - \rho \mathcal{V}_{(P,p)}(\xi_0^{\mathrm{EF}^{21}}, x_0; f). \tag{39}$$

Again, we say that for $(P, p) \in \mathbb{V}_3$, a Lyapunov $\mathcal{V}_{(P,p)}$ *satisfies rate* $\rho$ for the iterates of $\mathrm{EF}^{21}$, if we have that

$$0 \geqslant \sup_{\substack{d \in \mathbb{N} \\ f \in \mathcal{F}_{\mu,L} \\ x_0, x_\star \in \mathbb{R}^d}} \sigma_\rho^{\mathrm{EF}^{21}}(\xi_1^{\mathrm{EF}^{21}}, \xi_0^{\mathrm{EF}^{21}}; (P, p); f)$$

$$\text{s.t.} \quad (x_1, \xi_1^{\mathrm{EF}^{21}}) = \mathrm{EF}^{21}(x_0, \xi_0^{\mathrm{EF}^{21}}; f) \tag{40}$$

$$\nabla f(x_\star) = 0$$

Formally, we prove the following lemma.

**Lemma 2** ($\mathrm{EF}^{21}$ feasibility problem)**.** *Consider running Algorithm 3 with a deterministic compression operator satisfying Assumption 1 for some $\epsilon \in [0, 1]$ on any function satisfying Assumptions 2 and 3. There exists a nonzero candidate Lyapunov function $\mathcal{V}_{(P,p)}$ of the form defined in (38), satisfying a given rate $\rho > 0$, if and only if the following problem is feasible:*

$$\underset{\substack{P \in \mathbb{S}^3, \\ p \in \mathbb{R}, \\ \lambda_{ij} \geqslant 0, \\ \nu_i \geqslant 0}}{feasible} \begin{cases} 0 \succcurlyeq \Delta V_P(\rho) + \sum_{i,j \in \{0,1,\star\}} \lambda_{ij} M_{ij} + \nu \cdot C_i^{\mathrm{EF}^{21}} \\[2mm] 0 \geqslant_2 \Delta v_p(\rho) + \sum_{i,j \in \{0,1,\star\}} \lambda_{ij} m_{ij} \\[2mm] 0 \preccurlyeq P \\[1mm] 0 \leqslant p \\[1mm] 1 = \mathrm{Tr}(P) + p \end{cases} \tag{$\mathrm{EF}^{21}$-SDP}$$

*where matrices $(M_{ij})_{i,j \in \{0,1,\star\}}$ are defined as in Eq. (24), $(C_i^{\mathrm{EF}^{21}})_{i \in \{0,1\}}$ given below in (44), and $\Delta V_P(\rho), \Delta v_p(\rho)$ are given below in (45) and (46). Here $\geqslant_2$ denotes coordinate-wise inequality in $\mathbb{R}^2$.*

*Proof sketch.* We begin by changing our basis vectors to

$$\bar{x}_i := \mathbf{e}_{i+1}^\top, \bar{g}_i := \mathbf{e}_{i+3}^\top, \bar{c}_i := \mathbf{e}_{i+5}^\top, \bar{d}_i := \mathbf{e}_{i+7}^\top, \bar{f}_i := \mathbf{e}_i^\top, \quad i \in \{0, 1\} \tag{41}$$

where $\bar{x}_i, \bar{g}_i, \bar{c}_i, \bar{d}_i \in \mathbb{R}^8$, and $\bar{f}_i \in \mathbb{R}^2$. Similarly, we define the row vectors corresponding to the fixed-point as

$$\bar{x}_\star := \mathbf{0}_8^\top, \bar{g}_\star := \mathbf{0}_8^\top, \bar{c}_\star := \mathbf{0}_8^\top, \bar{d}_\star := \mathbf{0}_8^\top, \bar{f}_\star := \mathbf{0}_2^\top, \tag{42}$$

We define our method using these basis vectors as

$$\bar{x}_1 = \bar{x}_0 + \eta \cdot \bar{d}_0,$$
$$\bar{d}_1 = \bar{d}_0 + \bar{c}_0. \tag{43}$$

The only difference in the interpolation conditions is that we are compressing a different quantity now, which we can encode using

$$C^{\mathrm{EF}^{21}} := (\bar{g}_1 - \bar{d}_0 - \bar{c}_0)^\top (\bar{g}_1 - \bar{d}_0 - \bar{c}_0) - \epsilon(\bar{g}_1 - \bar{d}_0)^\top (\bar{g}_1 - \bar{d}_0). \tag{44}$$

where we only need a single matrix because the compression operator acts on a mixture of the current and next state.

Next, we encode the linear and quadratic terms in exactly the same way as we did for EF, except with the state variables

$$\Delta V_P(\rho) := \begin{bmatrix} \bar{x}_1 - \bar{x}_\star \\ \bar{g}_1 - \bar{g}_\star \\ \bar{d}_1 - \bar{d}_\star \end{bmatrix}^\top P \begin{bmatrix} \bar{x}_1 - \bar{x}_\star \\ \bar{g}_1 - \bar{g}_\star \\ \bar{d}_1 - \bar{d}_\star \end{bmatrix} - \rho \begin{bmatrix} \bar{x}_0 - \bar{x}_\star \\ \bar{g}_0 - \bar{g}_\star \\ \bar{d}_0 - \bar{d}_\star \end{bmatrix}^\top P \begin{bmatrix} \bar{x}_0 - \bar{x}_\star \\ \bar{g}_0 - \bar{g}_\star \\ \bar{d}_0 - \bar{d}_\star \end{bmatrix}. \tag{45}$$

$$\Delta v_p(\rho) := p\left(\bar{f}_1 - \bar{f}_\star\right) - \rho \cdot p\left(\bar{f}_0 - \bar{f}_\star\right), \tag{46}$$

Here too, $\Delta V_P(\rho) \in \mathbb{R}^{8 \times 8}$, and $\Delta v_p(\rho) \in \mathbb{R}^2$.

Finally, using **Step 1** and **Step 2** of Appendix A.1, we arrive at the feasibility problem described in the statement of the lemma. $\qquad \square$

## B    Rate comparison

In this section, we compare the rate given in equation (14) with the convergence rate of CGD and the standard convergence rate of EF$^{21}$ [14, see Theorem 2]. We show that CGD is strictly superior to EF/EF$^{21}$ using a CAS certificate, that our rate is strictly better than the existing rate for EF$^{21}$, and then we visualize the gap in both cases in Figures 2 and 4.

We also compare the "complexity" resulting from either method in Figures 3 and 5. In order to create a fair and interpretable comparison we used the following metric:

$$\frac{\log \rho_A}{\log \rho_B}. \tag{47}$$

The log-ratio can be motivated by the following example: if $\rho_A = 0.99$ and $\rho_B = 0.98$, then the latter will require half the number of iterations of the former before $\mathcal{V}(\xi_N, x_N; f) \leqslant \epsilon$.

### B.1    Comparison between CGD and EF/EF$^{21}$

We now compare the convergence rate when using the optimal step size between CGD and EF (equivalently, EF$^{21}$ since the rate is the same). Note that the optimal rate for CGD is given by [64, 65]

$$\rho_{\text{CGD}} = \left( \frac{\kappa_{\sqrt{\epsilon}} - 1}{\kappa_{\sqrt{\epsilon}} + 1} \right)^2, \tag{48}$$

where $\kappa_{\sqrt{\epsilon}} = \kappa \left( \frac{1 + \sqrt{\epsilon}}{1 - \sqrt{\epsilon}} \right)$, when using step size [64]

$$\eta_{\text{CGD}} = \frac{2}{(1 - \sqrt{\epsilon})\mu + (1 + \sqrt{\epsilon})L}. \tag{49}$$

The next statement has been verified using a WolframScript:

> **Certified using WolframScript**                                                                        **CAS** ↗
>
> $$\rho_{\text{EF}} - \rho_{\text{CGD}} = \sqrt{\epsilon} + \left( \frac{1 - \sqrt{\epsilon}}{2} \right) \left( \frac{\kappa - 1}{\kappa + 1} \right)^2 \left[ 1 - \sqrt{\epsilon} + \sqrt{(1 + \sqrt{\epsilon})^2 + \sqrt{\epsilon} 16 \frac{\kappa}{(\kappa - 1)^2}} \right]$$
> $$- \left( \frac{\kappa_{\sqrt{\epsilon}} - 1}{\kappa_{\sqrt{\epsilon}} + 1} \right)^2 > 0.$$

### B.2    Comparison to Richtárik et al. [14]

The rate of [14] for EF$^{21}$ under the Łojasiewicz inequality, and with the largest possible step size chosen, is given by

$$\rho' := \max \left\{ 1 - \frac{1 - \sqrt{\epsilon}}{2}, \ 1 - \kappa^{-1} \left( \frac{1 - \sqrt{\epsilon}}{1 + \sqrt{\epsilon}} \right) \right\} \tag{50}$$

in the single-agent case. We begin by noting that to show that our rate is faster, we only have to show that it is smaller than one of the terms of equation (50).

Note that

$$\rho' - \rho = 1 - \sqrt{\epsilon} - \frac{1 - \sqrt{\epsilon}}{(1 + \sqrt{\epsilon})\kappa}$$

$$- \frac{(1 - \sqrt{\epsilon})(\kappa - 1)^2 \left(1 - \sqrt{\epsilon} + \sqrt{1 + \epsilon + \frac{2\sqrt{\epsilon}\left(1 + \kappa(6 + \kappa)\right)}{(\kappa - 1)^2}}\right)}{2(\kappa + 1)^2}$$

Writing this as a single fraction, we see that the sign of this expression is the same as that of

$$\Delta := \left(1 - \sqrt{\epsilon}\right)\Big(-2 + \kappa\big(-3 + \epsilon(\kappa - 1)^2 + 2\sqrt{\epsilon}(\kappa + 1)^2 + (1 + \sqrt{\epsilon})R$$

$$+ \kappa\big(4 + \kappa - (1 + \sqrt{\epsilon})R\big)\big)\Big).$$

where $R := \sqrt{(\kappa - 1)^2 + \epsilon(\kappa - 1)^2 + 2\sqrt{\epsilon}\left(1 + \kappa(6 + \kappa)\right)}$. Denote the coefficient in front of $R$ as $C(R)$, and note that is negative. As a result, it just remains to make sure that the rest of the terms compensate for this term. Rewriting this as an inequality, and squaring to deal with $R$, our result follows from the inequality

$$(\Delta - C(R) \cdot R)^2 - (C(R) \cdot R)^2 = 4(1 - \sqrt{\epsilon})^2(1 + \kappa)F > 0,$$

where

$$F = 1 + \kappa\left(1 + \kappa(-5 + 3\kappa) + \epsilon(1 + \kappa)(-1 + 3\kappa) + 2\sqrt{\epsilon}\,(-1 + \kappa)(1 + 3\kappa)\right).$$

The factors other than $F$ are clearly positive, and F is increasing with respect to $\epsilon$. Setting $\epsilon$ to 0 shows that it is positive for all valid $\epsilon$.

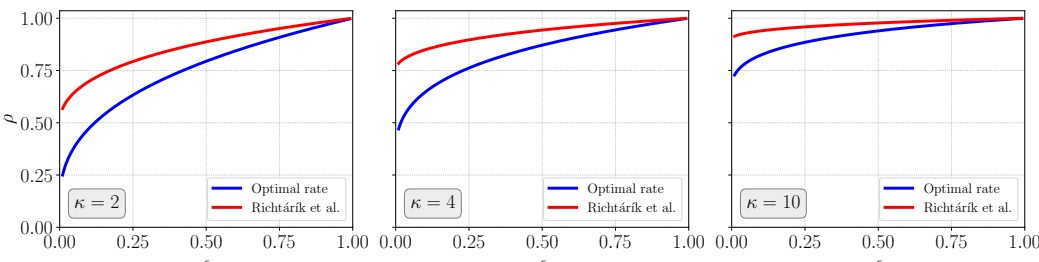

Figure 2: Line plot comparing the convergence rate of this paper (blue) with equation (50) (red) for various $\kappa$.

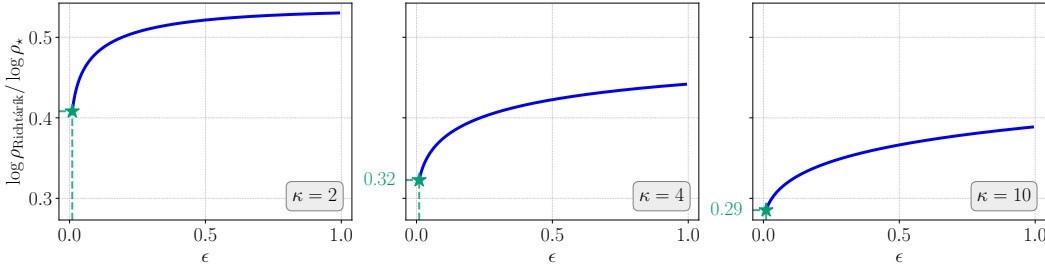

Figure 3: Line plot comparing the complexity of equation (50) with the rates of this paper for various $\kappa$.

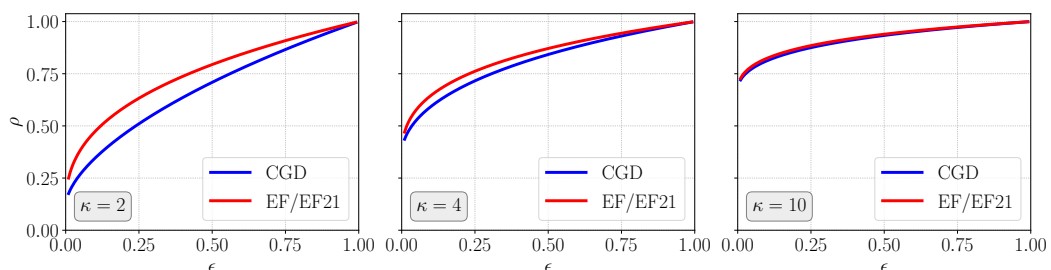

Figure 4: Line plot comparing the convergence rate of this CGD (blue) with EF/EF$^{21}$ (red) for various $\kappa$.

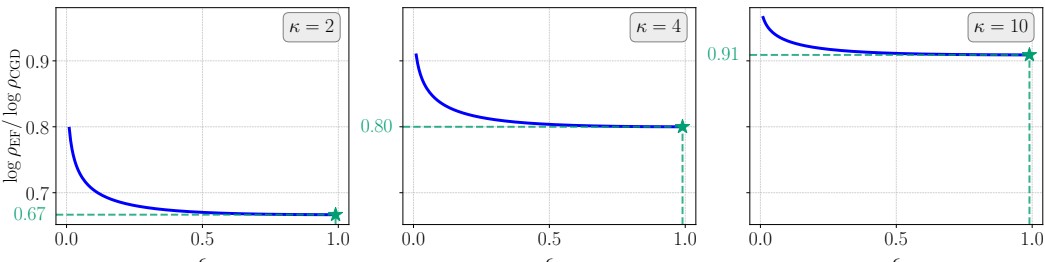

Figure 5: Line plot comparing the complexity of EF/EF$^{21}$ with CGD for various $\kappa$.

## C   Missing proofs

This section contains the proofs of Theorems 1 and 2. The proofs were obtained using by observing numerical results using the Lyapunov search procedure presented in Appendices A.2 and A.3. The resulting proofs are remarkably compact, and the main technical step consists of verifying an algebraic reformulation by hand. For ease of verification, we provide the reader with scripts written in Wolfram Language[4] that automatically performs those reformulations using a computer algebra system.

### C.1   Proof of Theorem 1

**Theorem 1.**

*Consider running Algorithm 2, i.e., EF, with a compression operator $\mathcal{C}$ satisfying Assumption 1 for some $\epsilon \in [0, 1]$ on any function satisfying Assumptions 2, and 3. Let the step size be given by*

$$\eta^\star = \left(\frac{2}{L+\mu}\right) \cdot \left(\frac{1-\sqrt{\epsilon}}{1+\sqrt{\epsilon}}\right). \tag{8}$$

*Then, we have that*

$$\rho^\star(\mathrm{EF}_{\eta^\star}) = \sqrt{\epsilon} + \tfrac{1}{4}(1+\sqrt{\epsilon})(L-\mu)\lambda, \tag{9}$$

*where*

$$\lambda := \tfrac{\eta^\star}{L+\mu}\left[(1-\sqrt{\epsilon})(L-\mu) + (1+\sqrt{\epsilon})\sqrt{(L-\mu)^2 + 16L\mu\tfrac{\sqrt{\epsilon}}{(1+\sqrt{\epsilon})^2}}\right]. \tag{10}$$

*A Lyapunov function achieving the rate in (9), with $\xi^{\mathrm{EF}}$ defined in (7), is given by*

$$\mathcal{V}(\xi^{\mathrm{EF}}, x; f) := \|x - x_\star\|^2 - 2(x - x_\star)^\top e + \left(1 + \frac{1}{\sqrt{\epsilon}}\right) \cdot \|e\|^2 = \|x - x_\star - e\|^2 + \frac{1}{\sqrt{\epsilon}}\|e\|^2, \tag{11}$$

*Finally, the step size in (8) is worst-case optimal for* EF: *$\forall \eta \geqslant 0$, we have $\rho^\star(\mathrm{EF}_\eta) \geqslant \rho^\star(\mathrm{EF}_{\eta^*})$.*

*Proof.* We begin by proving the rate given in (9) for our Lyapunov function. Consider the following inequalities, and associated with each of them the assigned multiplier [5]

$$
\begin{aligned}
I^{(1)}_{\mathcal{F}_{\mu,L}} &:= f(x_k) - f_\star - \nabla f(x_k)^\top (x_k - x_\star) + \frac{1}{2L}\|\nabla f(x_k)\|^2 \\
&\quad + \frac{\mu}{2(1-\mu/L)}\|x_k - x_\star - \frac{1}{L}\nabla f(x_k)\|^2 \leqslant 0, \qquad : \lambda
\end{aligned}
$$

$$I^{(2)}_{\mathcal{F}_{\mu,L}} := f_\star - f(x_k) + \frac{1}{2L}\|\nabla f(x_k)\|^2 + \frac{\mu}{2(1-\mu/L)}\|x_k - x_\star - \frac{1}{L}\nabla f(x_k)\|^2 \leqslant 0, \qquad : \lambda$$

$$I_{\mathcal{C}} := \|e_{k+1}\|^2 - \epsilon\|e_k + \eta\nabla f(x_k)\|^2 \leqslant 0, \qquad : \nu$$

where $\lambda$ is defined in (10), and $\nu := \frac{1}{\sqrt{\epsilon}}$.

> **Certified using WolframScript**                     **CAS ↗**
>
> Summing these inequalities with their multipliers, plugging in the update rules for $x_{k+1}$ and $e_{k+1}$, and using $\rho$ to denote the contraction factor we got in (9), we can rewrite the resulting inequality as:
>
> $$\rho \cdot \mathcal{V}(x_k, e_k) \geqslant \mathcal{V}(x_{k+1}, e_{k+1}) + a \cdot \|e_k - \frac{\rho-1}{a}(x_k - x_\star) + \frac{2(\sqrt{\epsilon}-1)}{a(L+\mu)}g_k\|^2, \tag{51}$$
>
> where
>
> $$a := (\rho - \sqrt{\epsilon}) \cdot \left(\frac{1+\sqrt{\epsilon}}{\sqrt{\epsilon}}\right). \tag{52}$$

---

[4]Wolfram Language is the programming language used in Mathematica. The scripts can be used to verify our rates without a paid license using Wolfram Engine.

[5]These multipliers correspond to closed forms for some of the variables of (EF-SDP) when the Lyapunov function is fixed as in the statement of the theorem.

The statement now follows from the simple inequality $\rho > \sqrt{\epsilon}$.

We now prove that the announced rate is tight. Consider the one-dimensional quadratic function

$$f_\mu(x) = \frac{\mu}{2}x^2. \tag{53}$$

The proof strategy used here is to show that the contraction for our Lyapunov function asymptotically matches the convergence rate announced in Theorem 1. We begin by fully exploiting Assumption 1 and set

$$c_k := \mathcal{C}(e_k + \eta\nabla f(x_k)) = (1 + \sqrt{\epsilon}) \cdot (\eta\nabla f(x_k) + e_k). \tag{54}$$

We can now rewrite the update rule for $x_{k+1}$ and $x_{k+2}$ to get an expression for $e_k$ and $e_{k+1}$ respectively:

$$e_k = \frac{1 - \eta\mu(1 + \sqrt{\epsilon})}{1 + \sqrt{\epsilon}}x_k - \frac{x_{k+1}}{1 + \sqrt{\epsilon}}, \quad e_{k+1} = \frac{1 - \eta\mu(1 + \sqrt{\epsilon})}{1 + \sqrt{\epsilon}}x_{k+1} - \frac{x_{k+2}}{1 + \sqrt{\epsilon}}, \tag{55}$$

after which we use the update rule for $e_{k+1}$ of Algorithm 2 to get a second-order recurrence relation for the sequence $\{x_k\}_{k=1}^{\infty}$:

$$\sqrt{\epsilon}x_k = x_{k+2} - (1 - \eta\mu - \sqrt{\epsilon}(1 + \eta\mu)) \cdot x_{k+1} \tag{56}$$

The solution to this recurrence relation is given by the roots of the characteristic equation, and after plugging in the initial conditions, we get

$$\begin{aligned} x_k = &\frac{1}{T} \cdot (1 - \eta\mu + \sqrt{\epsilon}(1 - \eta\mu) + T)(1 - \eta\mu - \sqrt{\epsilon}(1 + \eta\mu) + T)^k \\ &- \frac{1}{T} \cdot (1 - \eta\mu + \sqrt{\epsilon}(1 - \eta\mu) - T)(1 - \eta\mu - \sqrt{\epsilon}(1 + \eta\mu) - T)^k, \end{aligned} \tag{57}$$

where $T := \sqrt{4\sqrt{\epsilon} + (1 - \eta\mu + \sqrt{\epsilon}(1 + \eta\mu))^2}$. Note that for

$$\eta < \left(\frac{1}{\mu}\right) \cdot \left(\frac{1 - \sqrt{\epsilon}}{1 + \sqrt{\epsilon}}\right), \tag{58}$$

which is strictly larger than the step size given in (8), the above expression is dominated by the first term in the limit $k \to \infty$. If we plug in the resulting asymptotic expression for $x_k$ into the definition of $e_k$, and plug the resulting points into our Lyapunov function we get

$$\frac{\mathcal{V}(x_{k+1}, e_{k+1})}{\mathcal{V}(x_k, e_k)} \xrightarrow{k \to \infty} \frac{1}{4}(1 - \eta\mu - \sqrt{\epsilon}(1 + \eta\mu) + T)^2 \tag{59}$$

which, after plugging in the step size given in (8), is exactly the convergence rate announced in Theorem 1. The fact that our Lyapunov function is tight now follows from the remark made in Section 3.4.

Finally, we prove that the step size given in (8) is the optimal step size for our Lyapunov function. Note that the contraction factor

$$\rho(\eta) := \frac{1}{4}(1 - \eta\mu - \sqrt{\epsilon}(1 + \eta\mu) + T)^2, \tag{60}$$

that becomes the dominant term in the limit $k \to \infty$ of (59) is strictly decreasing in $\eta$. This is immediate from inspecting the sign of the derivative of the expression with respect to $\eta$:

$$\frac{d\rho(\eta)}{d\eta} = -\mu(1 + \sqrt{\epsilon})\frac{(1 - \eta\mu - \sqrt{\epsilon}(1 + \eta\mu) + T)^2}{2T}. \tag{61}$$

The rest of the proof now follows from instead considering the quadratic given by

$$f_L(x) := \frac{L}{2}x^2, \tag{62}$$

and repeating all the arguments stated above, except we instead consider step sizes

$$\eta > \left(\frac{1}{L}\right) \cdot \left(\frac{1 - \sqrt{\epsilon}}{1 + \sqrt{\epsilon}}\right), \tag{63}$$

and show that the contraction for our Lyapunov function is strictly decreasing for these step sizes. The argument now follows from the fact that the contraction factor on both of these quadratics are the same for the step size given in (8). $\square$

## C.2 Proof of Theorem 2

**Theorem 2.**

*Consider running Algorithm 3 with a compression operator satisfying Assumption 1 for some $\epsilon \in [0, 1]$ on any function satisfying Assumptions 2, and 3. Let the step size be given by $\eta^\star$ in (8). Then,*

$$\rho^\star(\mathrm{EF}^{21}{}_{\eta_\star}) = \rho^\star(\mathrm{EF}_{\eta_\star}). \tag{12}$$

*A Lyapunov function achieving the rate in (12) is given by*

$$\mathcal{V}(\xi^{\mathrm{EF}^{21}}, x; f) := (1 + \sqrt{\epsilon}) \cdot \|g\|^2 - 2g^\top d + \|d\|^2 = \|g - d\|^2 + \sqrt{\epsilon} \cdot \|g\|^2. \tag{13}$$

*Finally, the step size $\eta^\star$ is worst-case optimal for this algorithm.*

*Proof.* We begin by proving the rate given in (12) for our Lyapunov function. Consider the following inequalities, and associated with each of them the assigned multiplier:

$$I^{(1)}_{\mathcal{F}_{\mu,L}} := f(x_k) - f(x_{k+1}) + \frac{\|\nabla f(x_{k+1}) - \nabla f(x_k)\|^2}{2L} + \nabla f(x_k)^\top(x_{x+1} - x_k)$$
$$+ \frac{\mu}{2(1 - \mu/L)}\|x_k - x_{k+1} - \frac{1}{L}(\nabla f(x_k) - \nabla f(x_{k+1}))\|^2 \leqslant 0, \qquad : \lambda'$$

$$I^{(2)}_{\mathcal{F}_{\mu,L}} := f(x_{k+1}) - f(x_k) + \frac{\|\nabla f(x_k) - \nabla f(x_{k+1})\|^2}{2L} + \nabla f(x_{k+1})^\top(x_k - x_{k+1})$$
$$+ \frac{\mu}{2(1 - \mu/L)}\|x_{k+1} - x_k - \frac{1}{L}(\nabla f(x_{k+1}) - \nabla f(x_k))\|^2 \leqslant 0, \qquad : \lambda'$$

$$I_{\mathcal{C}} := \|\nabla f(x_{k+1}) - d_k - \mathcal{C}(\nabla f(x_{k+1}) - d_k)\|^2 - \epsilon\|\nabla f(x_{k+1}) - d_k\|^2 \leqslant 0, \qquad : \nu$$

where $\lambda'$ is defined as

$$\lambda' := \frac{\sqrt{\epsilon}}{\eta^\star(L + \mu)}\left[(1 - \sqrt{\epsilon})(L - \mu) + (1 + \sqrt{\epsilon})\sqrt{(L - \mu)^2 + \frac{16L\mu\sqrt{\epsilon}}{(1 + \sqrt{\epsilon})^2}}\right], \tag{64}$$

and $\nu := 1$.

> **Certified using WolframScript**      CAS ⧉
>
> Summing these inequalities with their multipliers, plugging in the update rules for $x_{k+1}$ and $d_{k+1}$, and using $\rho$ to denote the contraction factor we got in (12), we can rewrite the resulting inequality as:
>
> $$\rho \cdot \mathcal{V}(g_k, d_k) \geqslant \mathcal{V}(g_{k+1}, d_{k+1}) + a \cdot \|d_k + \frac{1}{a}((\epsilon + b)g_{k+1} - (\rho + b)g_k)\|^2, \tag{65}$$
>
> where
>
> $$b := \frac{\lambda}{L - \mu} \cdot \left(\frac{1 - \sqrt{\epsilon}}{1 + \sqrt{\epsilon}}\right). \tag{66}$$
>
> and
>
> $$a := \rho - \epsilon + 2\eta\lambda(1 - \sqrt{\epsilon})\frac{L\mu}{L - \mu}. \tag{67}$$

The statement now follows from plugging in the value of our multipliers and checking the sign.

We now prove that the announced rate is tight. Consider the one-dimensional quadratic function

$$f_\mu(x) = \frac{\mu}{2}x^2. \tag{68}$$

The proof strategy used here is to show that the contraction for our Lyapunov function asymptotically matches the convergence rate announced in Theorem 2. We begin by fully exploiting Assumption 1 and set

$$c_k := \mathcal{C}(\nabla f(x_{k+1}) - d_k) = (1 + \sqrt{\epsilon}) \cdot (\nabla f(x_{k+1}) - d_k) \tag{69}$$

We can now rewrite the update rule for $x_{k+1}$ and $x_{k+2}$ to get an expression for $d_k$ and $d_{k+1}$ respectively:

$$d_k = \frac{x_k - x_{k+1}}{\eta}, \quad d_{k+1} = \frac{x_{k+1} - x_{k+2}}{\eta}, \tag{70}$$

after which we use the update rule for $d_{k+1}$ of Algorithm 3 to get a second-order recurrence relation for the sequence $\{x_k\}_{k=1}^{\infty}$:

$$\sqrt{\epsilon} x_k = x_{k+2} - (1 - \eta\mu - \sqrt{\epsilon}(1 + \eta\mu)) \cdot x_{k+1}. \tag{71}$$

Note that this is the exact same recurrence relation as in (9), which means we can reuse the expression in (57) and the argument that follows. The proof now follows from the definition of our Lyapunov function in (13). The optimality of our Lyapunov function similarly follows from the same argument given in the remark made in Section 4.

Lastly, the proof of optimality of our step size follows directly from the same argument given in the proof of Theorem 1. $\qquad\square$

### C.3 Extension to stochastic compressors

Note that all the results of Theorems 1 and 2 were proven using the deterministic version of Assumption 1. Any deterministic compressor satisfies that assumption, and our lower bounds thus remain valid in this case. To show that the given rates are tight, we have to replace the inequalities $I_{\mathcal{C}}$ from the respective proofs with their stochastic counterparts:

$$I_{\mathcal{C}}^{\mathrm{EF}} := \mathbb{E}\left[\|e_{k+1}\|^2\right] - \epsilon\|e_k + \eta\nabla f(x_k)\|^2 \leqslant 0,$$

$$I_{\mathcal{C}}^{\mathrm{EF}^{21}} := \mathbb{E}\left[\|\nabla f(x_{k+1}) - d_k - \mathcal{C}(\nabla f(x_{k+1}) - d_k)\|^2\right] - \epsilon\|\nabla f(x_{k+1}) - d_k\|^2 \leqslant 0.$$

> **Certified using WolframScript**  CAS ↗
>
> For EF, we need to show the following:
>
> $$\rho \cdot \mathbb{E}\mathcal{V}(x_k, e_k) \geqslant \mathbb{E}\mathcal{V}(x_{k+1}, e_{k+1}),$$

> **Certified using WolframScript**  CAS ↗
>
> And for $\mathrm{EF}^{21}$ we need to show the following:
>
> $$\rho \cdot \mathbb{E}\mathcal{V}(g_k, d_k) \geqslant \mathbb{E}\mathcal{V}(g_{k+1}, d_{k+1}).$$

The same exact proof as used in the deterministic case holds for both due to the linearity of expectation. Wolfram Language scripts verifying this fact are available in the source code repository for this article.

# D  Additional numerical results

All subsections include details on how the corresponding results were computed. This section is organized as follows:

- Appendix D.1 presents additional performance plots for all methods and explains how the plots in the main paper were generated.

- Appendix D.2 provides illustrations demonstrating that our Lyapunov functions remain tight over multiple iterations.

- Appendix D.3 includes additional tables further confirming the tightness of our Lyapunov functions.

- Appendix D.4 shows plots illustrating the optimality of our step size.

- Appendix D.5 numerically demonstrates that the type of Lyapunov function used in Richtárik et al. [14] achieves a worse rate of convergence than the one used in this work.

All experiments were run on a MacBook Pro with an M4 Max processor. While none of the experiments are computationally demanding by modern standards, they can be scaled by increasing the resolution of the $\eta$ and $\epsilon$ grid to produce finer plots.

The source code for all the experiments is publically available in the following GitHub repository: https://github.com/DanielBergThomsen/error-feedback-tight

## D.1  Performance plots

This section presents the worst-case performance of all methods studied in this work, plotted as a function of the step size $\eta$ and the compression parameter $\epsilon$.

All contour plots were evaluated over a grid with $\epsilon \in [0.01, 0.99]$, and $\eta \in [0.01, \frac{2}{L+\mu_\star}]$, where $\mu_\star$ is the smallest $\mu$ specified in the caption of each figure (except for Figure 7, where it is set to 0.1). Each axis was discretized with a resolution of 200 points.

To generate each non-cyclic point, we used the following procedure:

1. For each method, we computed the optimal Lyapunov function (without additional constraints) via bisection on the contraction factor $\rho$, up to a precision of $10^{-6}$.

2. Using the resulting Lyapunov function, we then computed the worst-case contraction factor using PEPit [58] and the MOSEK solver [57].

We adopt this two-step approach because the feasibility problems used to compute the Lyapunov functions suffer from numerical instability. By evaluating the contraction factor separately using PEPit, we ensure that the reported value is an upper bound on the true contraction factor—up to solver tolerance.

To identify the area of non-convergence in the plots, we check whether a cycle exists for each pair of $\eta$ and $\epsilon$. This is done by following the procedure outlined in Goujaud et al. [66]: we compute the worst-case performance of the metric $-\|x_k - x_0\|^2$ for CGD, $-\|x_k - x_0\|^2 - \|e_k - e_0\|^2$ for EF, and $-\|x_k - x_0\|^2 - \|d_k - d_0\|^2$ for EF$^{21}$. If this value falls below a threshold (set to $10^{-3}$), we conclude that a cycle is present. In our experiments, cycles of length 2 were successfully identified for all methods, and these matched precisely with the regions of the contour plots where $\rho > 1$.

## D.2  Multi-step Lyapunov analysis

In this section, we show that our simple Lyapunov functions achieve the claimed convergence rate over multiple iterations. Specifically, we use PEPit to compute the contraction factor achieved by the Lyapunov function after $k$ iterations and compare it to the theoretical rate $\rho^k$, where $\rho$ is the single-step contraction factor. The exact match between these quantities in Figures 9 and 10 confirms that our single-step analysis accurately characterizes the worst-case performance over multiple iterations on these Lyapunov functions.

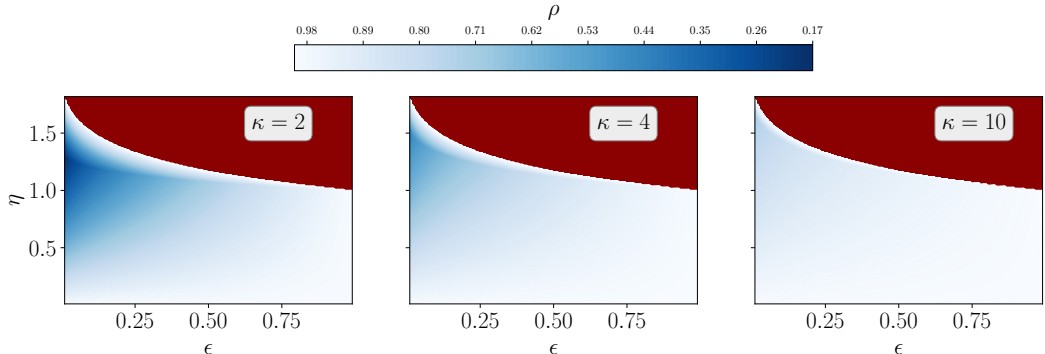

Figure 6: Contour plot showing the performance of CGD as a function of step size $\eta$ and compression parameter $\epsilon$, with regions of non-convergence marked in red. The regions of non-convergence were identified using PEPit by finding cycles of length 2. Each column corresponds to $\mu = 0.5, 0.25, 0.1$, with $L = 1.0$ fixed across all plots.

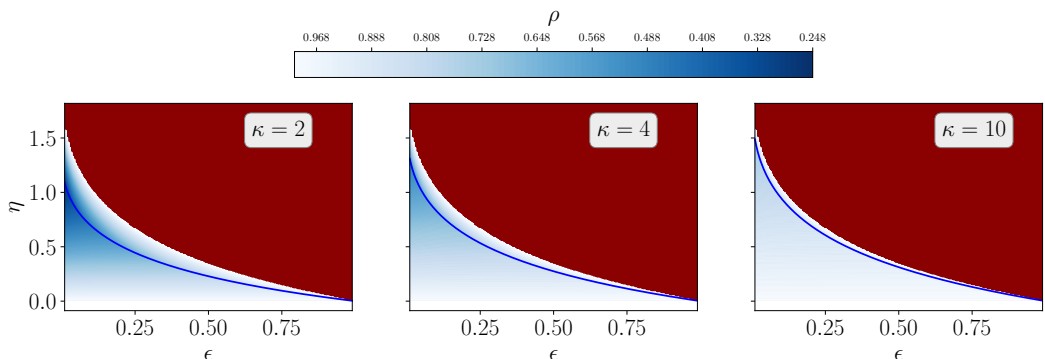

Figure 7: Contour plot showing the performance of EF as a function of step size $\eta$ and compression parameter $\epsilon$, with regions of non-convergence marked in red. The regions of non-convergence were identified using PEPit by finding cycles of length 2. Each column corresponds to $\mu = 0.5, 0.25, 0.1$, with $L = 1.0$ fixed across all plots. The optimal step size setting for a given $\epsilon$ is marked in blue.

### D.3 Lyapunov function class tightness

In this section, we show that for various conditioning numbers $\kappa$, our Lyapunov functions for EF and EF[21] are tight with respect to our class of Lyapunov functions, when using optimal step sizes. We remark that our Lyapunov functions are actually tight for many step size settings, but notably not step sizes which are larger than the optimal step size.

The tables report the maximum absolute difference in contraction factors between our Lyapunov function and the optimal one, over a range of $\epsilon$ and $\eta$ values specified in the captions. All contraction factors were computed using PEPit, and the procedure for the uncontrained Lyapunov functions is the one outlined in Appendix D.1. Points where either Lyapunov function yields a contraction factor greater than 1 were excluded from the computation of the maximum absolute difference.

|  | $\kappa = 2$ | $\kappa = 4$ | $\kappa = 10$ |
|---|---|---|---|
| Absolute error | 3.70e-07 | 4.83e-07 | 6.60e-07 |

Table 3: Maximum absolute difference in contraction factor for EF when comparing the general Lyapunov function—constructed using any combination of state terms specified in Subsection 3.1—to the simplified Lyapunov function defined in Theorem 1. The results are computed over a line with $\epsilon \in [0.01, 0.99]$ and with $\eta$ set to the optimal step size for $L = 1$, and $\mu = 0.5, 0.25, 0.1$.

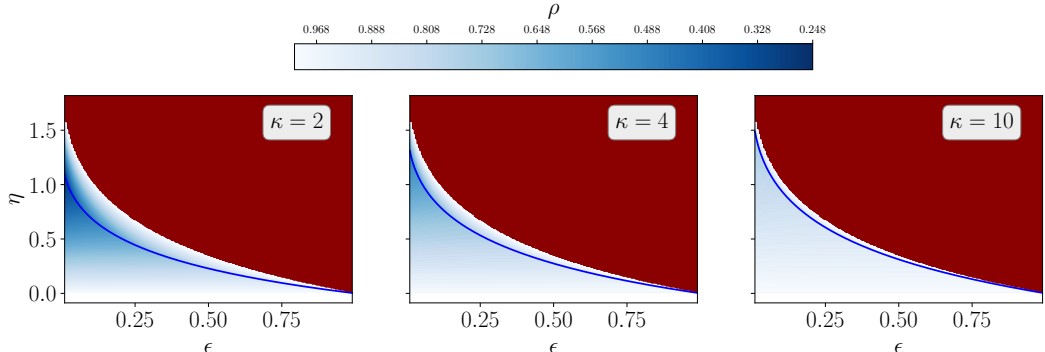

Figure 8: Contour plot showing the performance of $EF^{21}$ as a function of step size $\eta$ and compression parameter $\epsilon$, with regions of non-convergence marked in red. The regions of non-convergence were identified using PEPit by finding cycles of length 2. Each column corresponds to $\mu = 0.5, 0.25, 0.1$, with $L = 1.0$ fixed across all plots. The optimal step size setting for a given $\epsilon$ is marked in blue.

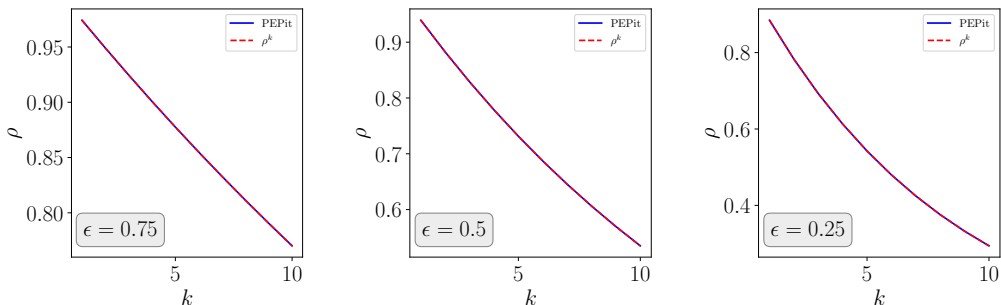

Figure 9: Multi-step Lyapunov analysis for EF, computed using PEPit. The blue line shows the contraction factor achieved by the Lyapunov function after $k$ iterations, while the red dashed line represents the theoretical rate $\rho^k$, where $\rho$ is the single-step contraction factor. Each column corresponds to a different value of $\epsilon = 0.75, 0.5, 0.25$, with $L = 1.0$ and $\mu = 0.1$ fixed across all plots.

|  | $\kappa = 2$ | $\kappa = 4$ | $\kappa = 10$ |
|---|---|---|---|
| Absolute error | 2.77e-07 | 1.97e-06 | 1.65e-06 |

Table 4: Maximum absolute difference in contraction factor for $EF^{21}$ when comparing the general Lyapunov function—constructed using any combination of state terms specified in Subsection 3.1—to the simplified Lyapunov function defined in Theorem 2. The results are computed over a line with $\epsilon \in [0.01, 0.99]$ and with $\eta$ set to the optimal step size for $L = 1$, and $\mu = 0.5, 0.25, 0.1$.

## D.4 Step size comparison

In this section, we compare the theoretically optimal step sizes we propose for our methods with empirically optimal step sizes determined through numerical experiments. To compute the empirical optima, we evaluate a grid of $\eta$ and $\epsilon$ values and select the step size that minimizes the contraction factor achieved by our simplified Lyapunov functions. The results of that experiment are found in Figure 11.

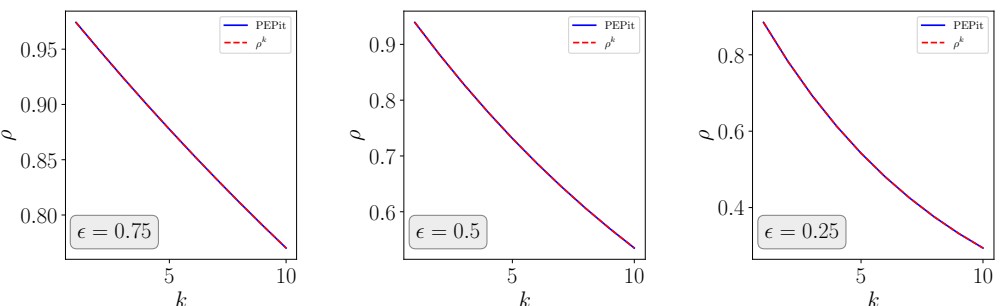

Figure 10: Multi-step Lyapunov analysis for $EF^{21}$, computed using PEPit. The blue line shows the contraction factor achieved by the Lyapunov function after $k$ iterations, while the red dashed line represents the theoretical rate $\rho^k$, where $\rho$ is the single-step contraction factor. Each column corresponds to a different value of $\epsilon = 0.75, 0.5, 0.25$, with $L = 1.0$ and $\mu = 0.1$ fixed across all plots.

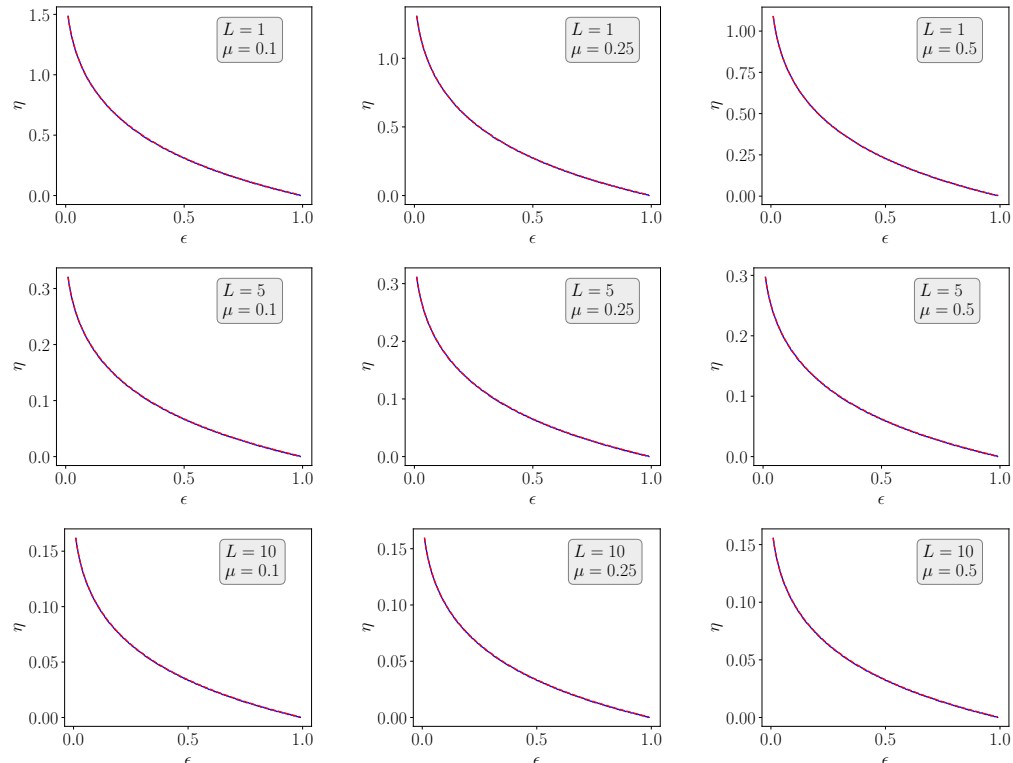

Figure 11: Empirically observed optimal step sizes (blue) in comparison with our setting (red, dashed) as a function of $\epsilon$ for different values of $\mu$ and $L$.

### D.5 Comparison of Lyapunov functions to previous work

In this subsection, we compare the convergence rates achieved with our Lyapunov function compared to that of Richtárik et al. [14]. For reference, the Lyapunov function they use is the following:

$$\mathcal{V}(\xi^{\mathrm{EF}^{21}}, x; f) := f(x) - f_\star + \frac{\eta}{\theta}\|d - g\|^2, \tag{72}$$

where $\theta := \frac{\epsilon}{1 - \sqrt{\epsilon}}$. We ran an experiment where we measured the best possible contraction factor $\rho'$ resulting form this parameterization, and compared it to the ones we achieve with the Lyapunov function defined in Theorem 2. In order to account for the possibility that there may be a better

weighting between the two terms of equation (72), we let that the weighting be a free parameter in the Lyapunov analysis PEP.

The metric used is the complexity metric introduced in the beginning of Appendix B. The result of this comparison can be found in Figure 12.

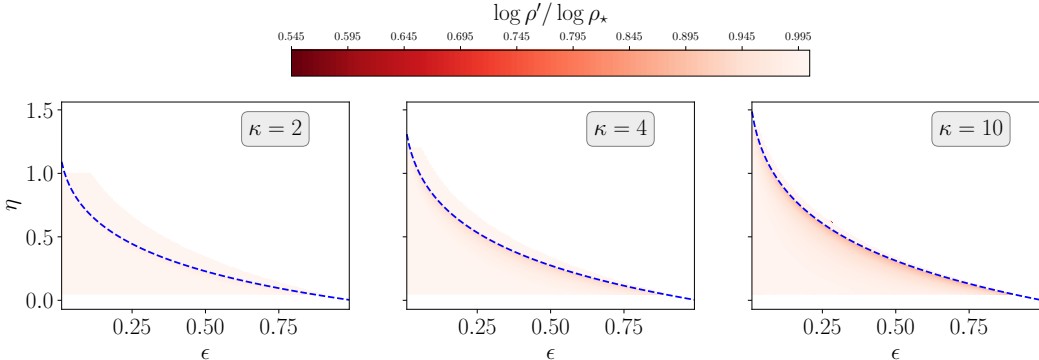

Figure 12: Complexity ratio between the rate achieved by the freely weighted version of equation (72), and the Lyapunov function used in Theorem 2.

# E   Remarks on the importance of proof certificates

The paper and appendix provide machine-checkable *certificates* for the main claims in the paper. We include two such complementary certificate types:

- [CAS] **Symbolic certificates (WolframScript).** Short WolframScript snippets that verify algebraic identities and intermediary equalities used within proofs. Any suitable Computer Algebra System (CAS) could be used for these purposes.

- [PEP] **Numerical certificates (PEPs).** Instances of the PEPs (as defined in the main text) that validate global inequalities and performance bounds by directly solving the corresponding optimization problems.

While the proofs resulting from PEP formulations are well suited to CAS for local verification of individual steps, we use both tools: WolframScript checks the algebraic steps, and PEPs validate the global statements.

The rationale behind the decision of adding these proof certificates is the following:

- **Reviewing efficiency and literature consolidation.** Submission volumes at large ML venues continue to grow, increasing reviewer workload[6] and raising challenges in the community, as acceptance of a paper does not necessarily guarantee that proofs have been thoroughly checked [7].

  While machine-checkable certificates do not replace mathematical proofs, they may shift effort from re-deriving statements to verifying that appropriate checks were provided, which is typically easier. They also provide authors with early feedback on correctness before submission.

- **Incremental path to formalization.** General-purpose proof assistants (e.g., Lean, Rocq) do not yet provide all mathematical infrastructure needed to formalize many ML results. As an interim step, we certify the parts that *are* currently amenable: symbolic steps via computer algebra and end-to-end statements via PEPs. This may enable a gradual transition toward fully formal proofs as libraries mature.

- **Clarifying experimental intent.** Ultimately, we also believe that this can serve to clarify the role of experiments in the papers, that are not always described. Some experiments test hypotheses; while others function as *theory unit tests*. Declaring the latter explicitly (and supplying their certificates) helps readers understand how these checks support the claims.

Formalization in Lean or Rocq is our ultimate standard for a guaranteed, machine-checked certificate of validity. We consider this an ambition for future work and aim to explore it by gradually linking the symbolic components to Lean (e.g., turning recurring identities into lemmas) and packaging numerical outputs as checkable witnesses.

---

[6] https://blog.neurips.cc/category/2024-conference/

[7] The NeurIPS 2025 reviewer guidelines indicate specifically that the reviewer is *not* required to read the paper's supplementary material https://neurips.cc/Conferences/2025/ReviewerGuidelines.

