# OpenReview forum: "Tight analyses of first-order methods with error feedback"
_NeurIPS.cc/2025/Conference — NeurIPS 2025 poster_

### Official Review · Reviewer_z1Gw · 2025-07-02

**Clarity:** 4
**Significance:** 2
**Originality:** 4
**Rating:** 5
**Confidence:** 4

**Summary:**

The paper provides a tight analysis of CGD, EF, and EF21 through the lens of the meta-optimization problem defined in equation (6). The authors detail the formulation of this meta-optimization problem, specify the set from which the Lyapunov function is selected, and explain the procedure used to construct the optimal Lyapunov function (i.e., how the meta-optimization problem is solved). The formal proofs of the convergences have been presented.

**Questions:**

1. How do the convergence rates obtained in this work compare to the classical results for EF and EF21?
2. What's the primary challenge of analyzing the distributed setting?

**Ethical Concerns:**

["NO or VERY MINOR ethics concerns only"]

**Final Justification:**

The main criticism of this paper concerns not the obtained results, but their relevance. While the analysis of the multi-node case is of utmost importance, the authors claim to have identified new properties in the single-node case, which appear to be relevant.

**Limitations:**

yes

**Paper Formatting Concerns:**

No formatting concerns

**Quality:**

4

**Strengths And Weaknesses:**

Strengths:
1. The paper addresses a compelling research question and utilizes an interesting tool to tackle it.
2. The theoretical analysis is rigorous, with carefully derived proofs.

Weaknesses:
1. The analysis is restricted to the single-client setting. This significantly limits the impact of the results, as some of the known challenges of CGD in the distributed setting, when $n > 1$, are not present in the $n=1$ case. In particular, as noted by the authors and shown in prior work \[1, 2], CGD fails to converge in the distributed setting. Therefore, the phrasing in line 239  “These results challenge the prevailing intuition that error feedback ensures convergence comparable to that of uncompressed methods” is misleading: the lack of convergence is not a matter of intuition but a rigorously established fact. On the other hand, CGD converges without issue in the single-node case. Additionally, in the single-node case, error feedback appears unnecessary and overly complex.
2. The paper lacks a clear rate comparison with classical results for EF [3] and EF21 [4]. In particular, Theorems 1 and 2 do not explicitly state the iteration complexity rates, and require additional manipulation to extract them.

References:
[1] Beznosikov et al, "On Biased Compression for Distributed Learning"

[2] Karimireddy et al, "Error Feedback Fixes SignSGD and other Gradient Compression Schemes"

[3] Stich et al, "SGD with memory"

[4] Richtarik et al, "EF21: Error Feedback 2021"

---

> ### Author Rebuttal · Authors · 2025-07-31
>
> We sincerely thank the reviewer for their positive feedback and high ratings regarding the quality, clarity, and originality of our work. We also appreciate their familiarity with the relevant literature. Below, we respond to each of the reviewer’s comments and questions in detail.
>
> >The analysis is restricted to the single-client setting. This significantly limits the impact of the results, as some of the known challenges of CGD in the distributed setting, when n>1, are not present in the n=1 case. In particular, as noted by the authors and shown in prior work [...], CGD fails to converge in the distributed setting.
>
> > Additionally, in the single-node case, error feedback appears unnecessary and overly complex.
>
> While we all agree that the multi-agent case is of importance and shows nuances, we believe the single-agent case to be of significant interest, and do not fully agree with the claim that EF is "unnecessary and overly complex" in the single-agent setting.
>
> 1. Many foundational works on error feedback [1–3] perform their analysis in the single-agent setting. E.g. Table 1 in [5] highlights this. It thus seems that the community does not disregard the importance of this regime for theoretical understanding.
>
> 2. We believe our analysis does reveal meaningful distinctions:
> - CGD does not consistently outperform EF or EF21 across all step sizes. The superiority of CGD only holds under optimal tuning. In fact, one of CGD’s key advantages is its ability to operate using significantly larger step sizes, which offers useful intuition for its behavior and performance trade-offs. We generated a figure (prior to the announcement disallowing new figures in the rebuttal process) that showcases, for any fixed $(\eta, \epsilon)$, the log-ratio of optimization complexities between CGD and EF: for instance, for $\kappa=10$, depending on the step size, EF can be more than 100 times faster or 1000 times slower than CGD (in terms of number of steps required to achieve the same reduction in performance measure). We will add this figure to the paper.
>
> - The equality of rates between EF and EF21 in this setting was, to the best of our knowledge, previously unknown. This equivalence can be revealed only by analysis using an optimal Lyapunov function, as the optimal rate is achieved for a very different Lyapunov function. For a fixed Lyapunov function, the rates of EF and EF21 are *not* the same—we will also provide an illustration of this fact.
>
> 3. Third, there exist links to quantized-aware-training of deep neural networks, which we should have highlighted (and will add):
>
> BinaryConnect [6], which is one of the most popular algorithms for *quantization-aware training* of neural networks, combines gradient steps with quantization steps. Although not directly equivalent to CGD or EF, it is a natural example of a **single-agent** algorithm that combines quantization and GD. The adaptation of EF in the context of sparse neural networks leads to the work of Lin et al. [7], who perform sparse neural-network training by applying only sparse updates, which is equivalent to error feedback. The authors note this equivalence at the end of Section 3 (and are themselves among the active contributors to the field of error feedback, as they have co-authored [1-4]).
>
> **Generally speaking, the single-agent setting is thus relevant to the design of training methods for sparse or quantized neural networks**—motivating the study of these algorithms even outside their typical distributed setting.
>
>
>
> - [1] S. Stich, S.P. Karimireddy. *The error-feedback framework: Better rates for SGD with delayed gradients and compressed communication.* JMLR, 2020.
> - [2] A. Ajalloeian, S. Stich. *Analysis of SGD with biased gradient estimators.* , 2020.
> - [3] S.P. Karimireddy, Q. Rebjock, S. Stich, M. Jaggi. *Error feedback fixes SignSGD and other gradient compression schemes*. ICML, 2019.
> - [4] S. Stich, J-B. Cordonnier, M. Jaggi. *Sparsified SGD with memory*. Neurips 2018.
> - [5] P. Richtárik, I. Sokolov, I. Fatkhullin. *EF21: A new, simpler, theoretically better, and practically faster error feedback*. Neurips, 2021.
> - [6] M. Courbariaux, Y. Bengio, J-P. David. *BinaryConnect: Training Deep Neural Networks with binary weights during propagations*. Neurips 2015.
> - [7] T. Lin, S. Stich, L. Barba, D. Dmitriev, M. Jaggi. *Dynamic Model Pruning with Feedback*. ICLR, 2020.
>
>
> > Therefore, the phrasing in line 239 “These results challenge the prevailing intuition that error feedback ensures convergence comparable to that of uncompressed methods” is misleading: the lack of convergence is not a matter of intuition but a rigorously established fact. On the other hand, CGD converges without issue in the single-node case.
>
> There might a misunderstanding here: the claim we make in line 239 (and that echoes the way methods using error feedback are sometimes understood by the community) is that methods using error feedback achieve rates of convergence almost equivalent to the "no-compression case" (uncompressed methods), not to the "CGD case". What we show is that in the single agent deterministic case, they neither recover the performance of GD, nor that of CGD.  More precisely, CGD, which we consider to be the baseline method in the single-agent setting, is considerably slower than uncompressed methods, and as EF and EF21 underperform relative to the baseline method in the single-agent case, they are by-extension also not comparable to uncompressed methods.
>
> We thus think the phrasing of line 239 is fair, but can amend it if the reviewer believes it's misleading.
>
> >The paper lacks a clear rate comparison with classical results for EF [3] and EF21 [4]. In particular, Theorems 1 and 2 do not explicitly state the iteration complexity rates, and require additional manipulation to extract them.
>
> >Q: How do the convergence rates obtained in this work compare to the classical results for EF and EF21?
>
> We agree that our original presentation did not make rate comparison easy. We highlighted the dependence on the step-size and the optimal Lagrange multiplier, but the resulting expression is not immediately comparable to existing bounds. For completeness, we provide below the cleanest closed form we have derived to date for our rate:
> $$
> \rho =
> \sqrt{\epsilon} + \left(\frac{1 - \sqrt{\epsilon}}{2}\right) \left(\frac{\kappa - 1}{\kappa + 1}\right)^2 \left[1 - \sqrt{\epsilon} + \sqrt{(1 + \sqrt{\epsilon})^2 + \sqrt{\epsilon} 16 \frac{\kappa}{(\kappa - 1)^2}}\right],
> $$
> where $\kappa := L/\mu$. Showing that this bound strictly improves upon other existing rates, such as the one under the PL-condition rate of [1], is straightforward (and natural: we have a tight rate under stricter assumptions); quantifying the margin analytically is less so, though feasible. We will therefore include the above parameterization as a remark in the revision, together with both an analytical discussion and a visual comparison against the PL-based rate.
>
> >Q: What's the primary challenge of analyzing the distributed setting?
>
> This is a very good question. The main bottleneck in extending our approach to the multi-agent case is that the candidate Lyapunov function space grows quadratically (the Lyapunov functions may contain gradients, compressed messages from different functions, and error feedback terms, as well as crossed inner products); should the optimal function involve dense coefficient patterns, obtaining a closed-form via our approach requires substantial additional effort.
> One approach in that direction, that we are currently investigating, is the use of symmetries to reduce the size of the problem.
>
> ---
> We thank the reviewer for their support and remain available for any further discussion.
>
> ---

---

> > ### Comment · Reviewer_z1Gw · 2025-08-06
> >
> > Thank you for addressing my questions in the rebuttal.
> >
> > **Single-node case analysis**
> >
> > 1. I agree that the single-node analysis, at least, was valuable to the community. A few years ago, we lacked a solid understanding of error feedback -- both in terms of its convergence and underlying intuition -- and single-node analysis helped move the field forward. However, while some open questions persist, many now lie in the distributed setting, which arguably deserves more attention. Notably, the key references cited by the authors ([1–3]) date back to 2020, highlighting that the foundational single-node work is already a few years old.
> >
> > 2. Thank you for your remark regarding CGD's superiority. I look forward to examining the referenced plot in the final version. As for the equivalence between EF and EF21, I initially recalled the section *“Restricted Equivalence of EF and EF21”* in [1], but upon re-reading, I see that the assumptions required for the equivalence are indeed quite restrictive. Moreover, the equivalence is shown at the algorithmic level, not in terms of convergence rates. In that light, the result in this paper appears novel and meaningful.
> >
> > 3. Thank you for pointing me to [2]. While it's an interesting direction, I believe that the use of fixed masks versus contractive compressors leads to fundamentally different algorithmic behavior. This distinction is reflected in the convergence theorems: EF achieves exact convergence, whereas the method in [2] only converges to a neighborhood, which is expected given the fixed pruning mask.
> >
> > **Phrasing**
> >
> > Thank you for the clarification. I had misread the sentence.
> >
> > **The rate formula**
> >
> > I appreciate the authors providing the formula explicitly. While it is indeed quite long, including it in the final version would be valuable.
> >
> > ---
> >
> > **In conclusion**, I believe the paper merits acceptance and will update my score accordingly.
> >
> > ---
> >
> > References:
> > [1] P. Richtárik, I. Sokolov, I. Fatkhullin. EF21: A new, simpler, theoretically better, and practically faster error feedback. Advances in Neural Information Processing Systems 34, 2021.
> > [2] T. Lin, S. Stich, L. Barba, D. Dmitriev, M. Jaggi. Dynamic Model Pruning with Feedback. ICLR, 2020.

---

> > > ### Author Response · Authors · 2025-08-06
> > >
> > > Thank you very much for your feedback and support!

---

### Official Review · Reviewer_sQ6n · 2025-07-03

**Clarity:** 4
**Significance:** 2
**Originality:** 3
**Rating:** 5
**Confidence:** 4

**Summary:**

The paper provides tight bounds for Compressed GD, EF, and EF21. It employs the performance estimation framework to find the optimal Lyapunov functions for EF and EF21. The paper demonstrates that CSG actually converges for the larger set of problem parameters compared with EF and EF21.

**Questions:**

Minor concerns:

Line 188: redundant “and”

Line 220: better to move the reference to the end of the phrase

Line 237: naemly

Line 565 is strange

**Ethical Concerns:**

["NO or VERY MINOR ethics concerns only"]

**Limitations:**

yes

**Quality:**

4

**Strengths And Weaknesses:**

The paper is very well-written and the main technique for finding optimal Lyapunov functions was very interesting.

Concerns:

The paper only considers the exact-gradient case. Note that EF was largely motivated by the need of compressed GD to handle the stochastic noise (see [“Error Feedback Fixes SignSGD and other Gradient Compression Schemes”](https://arxiv.org/pdf/1901.09847), Section 3). Since the practical scenarios have stochastic noise, I question the significance of this work, in particular the conclusion “We also observed that their performance is strictly worse than that of compressed gradient descent – an outcome that, we believe, challenges the intuition of many in the field.”

Section 3: what is the optimal convergence rate for CGD?

In Section 2.1, you don’t talk about EF. Also, please mention EF21 when using reference [14].

---

> ### Author Rebuttal · Authors · 2025-07-30
>
> We thank the reviewer for their very positive comments on the execution of the paper, and for the ratings given. The typos found by the reviewer will be fixed in the next draft, and we thank them for pointing those out as well.
>
> >The paper only considers the exact-gradient case. Note that EF was largely motivated by the need of compressed GD to handle the stochastic noise (see [“Error Feedback Fixes SignSGD and other Gradient Compression Schemes”](https://arxiv.org/pdf/1901.09847), Section 3). Since the practical scenarios have stochastic noise, I question the significance of this work, in particular the conclusion “We also observed that their performance is strictly worse than that of compressed gradient descent – an outcome that, we believe, challenges the intuition of many in the field.”
>
> We agree with the reviewer that it would be very interesting to see what the guarantees would look like with stochastic gradients. That being said, it would change the scope of the paper quite a bit. Deterministic gradients have been used in some landmark papers (e.g. [1]), or it is reported as a separate result  (e.g. [2]). We consider it to be a different problem that we would prefer to leave as an exciting direction for future work.
>
> >Section 3: what is the optimal convergence rate for CGD?
>
> We recover the following optimal rate for CGD, that was also given by [3, 4]:
> $$
> \left(\frac{\kappa_{\sqrt{\epsilon}} - 1}{\kappa_{\sqrt{\epsilon}} + 1} \right)^2,
> $$
> where $\kappa_{\sqrt{\epsilon}} = \frac{L (1 + \sqrt \epsilon)}{\mu (1 - \sqrt \epsilon)}$ (note that we define the conditioning number inversely compared to [3]).
>
> The line for CGD in Figure 2 was generated simply by discretizing over the interval $\eta \in (0, 2/(L+\mu)]$, and manually selecting the step size that gave the best contraction. To reassure the reviewer: the plot was originally (before we knew the closed-form optimal step size for EF and EF21) using the same implementation, and simply looked like a discrete approximation of the smooth figure now present in the paper. We will update the plot to instead use the step size given in [4] directly in the next draft (while providing the reference and closed-form explicitly in the text).
>
> >In Section 2.1, you don’t talk about EF. Also, please mention EF21 when using reference [14].
>
> We acknowledge that we should have discussed existing EF results and will incorporate this in the next draft.
> We will likewise replace the generic reference with an explicit mention of EF21 accompanied by the Richtárik citation.
>
> ---
>
> We thank the reviewer again for their support and would happily participate in any further discussion if needed.
>
> ---
>
>
> References:
> - [1] P. Richtárik, I. Sokolov, I. Fatkhullin. _EF21: A new, simpler, theoretically better, and practically faster error feedback_. Advances in Neural Information Processing Systems 34, 2021.
> - [2] K. Gruntkowska, A. Tyurin, P. Richtárik. _EF21-P and Friends: Improved Theoretical Communication Complexity for Distributed Optimization with Bidirectional Compression_. International Conference on Learning Representations, 2020.
> - [3] E. De Klerk, F. Glineur, A. Taylor. _On the worst-case complexity of the gradient method with exact line search for smooth strongly convex functions_. Optimization Letters, 2016.
> - [4] O. Gannot. *A frequency-domain analysis of inexact gradient methods*. Mathematical programming, 2020.

---

### Official Review · Reviewer_f2XL · 2025-07-03

**Clarity:** 3
**Significance:** 2
**Originality:** 2
**Rating:** 3
**Confidence:** 3

**Summary:**

In this work, the authors provide a tight analysis of both EF and EF21 methods. They identify the Lyapunov function that gives the best possible convergence rate for each method and, using computer-assisted techniques, establish matching lower bounds that confirm the optimality of these rates. This careful approach provides sharp performance guarantees and allows for a clear, direct comparison between EF, EF21, and Compressed Gradient Descent (CGD). The analysis is conducted in a simplified setting, focusing on the single node and deterministic case, which highlights the core algorithmic ideas and enables clean theoretical insights without the added complexity of other settings.

**Questions:**

Please check the weaknesses section.

I think the paper is rigorous and clear, but the issues raised in the weaknesses section are important.

**Ethical Concerns:**

["NO or VERY MINOR ethics concerns only"]

**Final Justification:**

Dear all,

I appreciate the authors' responses and clarifications.

My main comments are as follows:

1. I appreciate the authors for their tight analysis in the single-node setting with deterministic gradients under the strongly convex and Lipschitz smooth assumptions.

2. While I acknowledge certain conceptual similarities, I do not see a strong connection between quantization-aware training and error-feedback methods in the single-node setting, as the former typically involves compression of models/iterates.

3. I also appreciate the rigorous analysis and the use of carefully designed Lyapunov functions. However, I remain somewhat unconvinced that the insights derived in the single-node, deterministic setting can be directly generalized to more practical or broader scenarios.

4. As previously noted, I find the single-node deterministic setting rather limited in scope. In my view, extending the analysis to the multi-node case would significantly enhance the impact and relevance of the work.

Based on these points, I believe a borderline reject is an appropriate score for this paper.

Best regards,

Reviewer

**Limitations:**

The results are obtained only in the single-node case and the deterministic setting.

**Quality:**

3

**Strengths And Weaknesses:**

Strengths:

The paper is well written, carefully structured, and easy to follow in most parts. The authors present their ideas clearly and maintain a logical flow throughout the paper, which makes it accessible to the reader.

All theorems, assumptions, and lemmas are clearly stated and supported by intuitive explanations. The theoretical sections are presented in a way that is both rigorous and understandable, which I find valuable.

The background section and the discussion of related work are thorough. The authors provide a solid overview of the existing literature and clearly explain how their work connects to and differs from previous studies.

The pseudocode is well presented and easy to interpret. It offers a clear description of the algorithms, making the proposed methods easier to understand and potentially easier to implement.

Figures 1 and 2, as well as Tables 1 and 2, are well designed, clearly labeled, and well explained in the text. They effectively support the theoretical findings of the paper. I appreciate the clarity and care in the visual presentation.

I briefly checked the theoretical results. They appear to be correct and rigorous, although I might have missed some minor details.

Overall, the results seem to be sound and are well explained. The paper demonstrates careful work and contributes insights to the topic.


Weaknesses:

The abstract is confusing and potentially misleading, as it does not mention that the results are obtained in the single-node setting.

Assumption 1 is made for deterministic compressors, while in the EF21 paper by Richtárik, the contractive compression operator is defined for stochastic compressors in expectation. Is it possible to extend the current results to the class of stochastic contractive compressors?

In line 132, there might be a typo in the formula (though I am not certain). I believe it should be $
\Vert g - \nabla f(x) \Vert^2 \leq \epsilon \Vert \nabla f(x) \Vert^2.
$

Please clarify.

The main issue with the paper is its focus on the single-node regime. The core motivation for using gradient-type methods with compressed updates, such as EF, EF21, and CGD (Compressed Gradient Descent) is typically in the multi-node setting. It is unclear why it would be meaningful to consider these methods in the single-node setting, where compression is not useful. Could the authors clarify any potential benefits of using such methods in the single-node case?

The authors claim that studying the single-node setting can provide intuition. However, I find this potentially misleading. In the single-node setting, CGD can converge with an appropriate stepsize without any issues. In this scenario, EF and EF21 mechanisms seem redundant, since the standard CGD method can work without additional memory or algorithmic adjustments. On the other hand, in the multi-node setting, as the authors themselves mention, CGD may diverge regardless of the stepsize. This indicates that the two settings are fundamentally different. It is unclear how insights from the single-node case can reliably transfer to the multi-node case.

Another important point is that EF21 is typically analyzed in non-convex settings and also under the Polyak-Łojasiewicz (PL) condition. This gives EF21 a theoretical advantage in non-convex problems. The results provided in this paper focus on the strongly convex case and do not address the more interesting non-convex scenario. Showing that EF and EF21 perform identically in the strongly convex setting does not necessarily imply that the same will hold in the non-convex or multi-node case. In fact, theory currently favors EF21 in the non-convex regime.

Finally, another limitation is that the analysis assumes full-gradient updates. Stochasticity is not considered, even though stochastic gradients are essential in practical applications. Therefore, it would be valuable to compare these methods in the stochastic case.

---

> ### Author Rebuttal · Authors · 2025-07-30
>
> We sincerely thank the reviewer for their thoughtful and constructive feedback on our work. Below, we address each point in the order raised.
>
> > The abstract does not mention single-agent.
>
> We agree that the final clause of the abstract is ambiguous. We will revise it to:
> "Our analysis is carried out in **the simplified single-agent setting**, which allows..."
>
> > Assumption 1 is made for deterministic compressors, while in the EF21 paper by Richtárik, the contractive compression operator is defined for stochastic compressors in expectation. Is it possible to extend the current results to the class of stochastic contractive compressors?
>
> This is a good question: the proofs indeed carry over directly to stochastic contractive compressors—one simply applies the expectation bound on the compressor and uses linearity of expectation in the Lyapunov analysis. We have prepared and will provide a WolframScript that verifies the resulting inequalities, and we will spell out the stochastic version of the assumption as well as the corresponding theorem statements in the next revision. We thank the reviewer for making this comment, which helps broaden the scope of the analysis.
>
> > Typo l.132
>
> We thank the reviewer for noticing this typo. It will be fixed in the next draft.
>
> > The main issue with the paper is its focus on the single-node regime. The core motivation for using gradient-type methods with compressed updates, such as EF, EF21, and CGD (Compressed Gradient Descent), is typically in the multi-node setting. It is unclear why it would be meaningful to consider these methods in the single-node setting, where compression is not useful. Could the authors clarify any potential benefits of using such methods in the single-node case?
>
> > The authors claim that studying the single-node setting can provide intuition. However, I find this potentially misleading. In the single-node setting, CGD can converge with an appropriate step size without any issues. In this scenario, EF and EF21 mechanisms seem redundant, since the standard CGD method can work without additional memory or algorithmic adjustments. On the other hand, in the multi-node setting, as the authors themselves mention, CGD may diverge regardless of the step size. [...]
>
> We appreciate this important question—this is a point we should have made clearer and will detail in the revised version. We fully acknowledge (and we stated in the submitted draft) that some behaviors will differ when moving from the single-agent to the multi-agent setting. Nevertheless, we believe there are strong arguments in favor of providing a tight analysis in the single-agent case as a first step:
>
> **1. Practical relevance beyond distributed optimization**
>
> We would like to highlight the relevance of our work in the context of sparse/quantized neural-network training. BinaryConnect [1], which is one of the most popular algorithms for *quantization-aware training* of neural networks, combines gradient steps with quantization steps. Although not directly equivalent to CGD or EF, it is a natural example of a **single-agent** algorithm that combines quantization and GD. The adaptation of EF in the context of sparse neural networks leads to the work of Lin et al. [2], who perform sparse neural-network training by applying only sparse updates, which is equivalent to error feedback. The authors note this equivalence at the end of Section 3 (and are themselves among the active contributors to the field of error feedback, as they have co-authored [3-6]).
>
> Generally speaking, the single-agent setting is thus relevant to the design of training methods for sparse or quantized neural networks—motivating the study of these algorithms even outside their typical distributed setting.
>
> - [1] M. Courbariaux, Y. Bengio, J-P. David. *BinaryConnect: Training Deep Neural Networks with binary weights during propagations*. Neurips 2015.
> - [2] T. Lin, S. Stich, L. Barba, D. Dmitriev, M. Jaggi. *Dynamic Model Pruning with Feedback*. ICLR, 2020.
>
> **2. Numerous key papers derive their analysis in the single-node case**
>
> We respectfully disagree that error feedback is simply redundant in the single-agent setting. In fact, many of the foundational (and well-cited) works that analyze error feedback [see, among others, 3–5] place themselves specifically in that single-agent regime for the analysis. This is underlined in other papers; see, for example, Table 1 of [7].
> We believe such sustained interest means that the community considers the single-agent case insights to be valuable.
>
> - [3] S. Stich, S.P. Karimireddy. *The error-feedback framework: Better rates for SGD with delayed gradients and compressed communication.* JMLR, 2020.
> - [4] A. Ajalloeian, S. Stich. *Analysis of SGD with biased gradient estimators.* , 2020.
> - [5] S.P. Karimireddy, Q. Rebjock, S. Stich, M. Jaggi. *Error feedback fixes SignSGD and other gradient compression schemes*. ICML, 2019.
> - [6] S. Stich, J-B. Cordonnier, M. Jaggi. *Sparsified SGD with memory*. Neurips 2018.
> - [7] P. Richtárik, I. Sokolov, I. Fatkhullin. *EF21: A new, simpler, theoretically better, and practically faster error feedback*. Neurips, 2021.
>
> **3. Comparing EF, EF21, and CGD is far from straightforward even in the single-node case, and our analysis in the single-agent case already brings valuable insights:**
>
> - **First**, CGD does not consistently outperform EF or EF21 across all step sizes. The superiority of CGD only holds under optimal tuning. In fact, one of CGD’s key advantages is its ability to operate under significantly larger step sizes, which offers useful intuition for its behavior and performance trade-offs. We generated a figure (prior to the announcement disallowing new figures in the rebuttal process) that showcases, for any fixed $(\eta, \epsilon)$, the log-ratio of optimization complexities between CGD and EF: for instance, for $\kappa=10$, depending on the step size, EF can be more than 100 times faster or 1000 times slower than CGD (in terms of number of steps required to achieve the same reduction in performance measure). We will add this figure to the paper.
>
> - **Second**, the equality of rates between EF and EF21 in this setting was, to the best of our knowledge, previously unknown. This equivalence can be revealed only by analysis using an optimal Lyapunov function, as the optimal rate is achieved for a very different Lyapunov function. For a fixed Lyapunov function, the rates of EF and EF21 are *not* the same—we will also provide an illustration of this fact.
>
> We therefore believe that, although some behaviors will differ in the multi-agent case, the simplified setting considered in this work reveals non-trivial and meaningful insights. Providing a tight analysis under optimal step-size tuning allows us to offer a fair comparison between the methods and should act as a stepping stone for future work comparing these methods in the multi-agent, non-convex, or stochastic-gradient cases.
> Nevertheless, we are happy to better highlight the differences to be expected in multi agent in a dedicated subsection.
>
> > Another important point is that EF21 is typically analyzed in non-convex settings and also under the Polyak-Łojasiewicz (PL) condition. [...] . Showing that EF and EF21 perform identically in the strongly convex setting does not necessarily imply that the same will hold in the non-convex or multi-node case. In fact, theory currently favors EF21 in the non-convex regime.
>
> > Finally, another limitation is that the analysis assumes full-gradient updates. Stochasticity is not considered, even though stochastic gradients are essential in practical applications. Therefore, it would be valuable to compare these methods in the stochastic case.
>
> - **Regarding extensions**: We agree that our work does not provide direct results under non-convex assumptions, nor does it use stochastic gradients. To avoid confusion, we have explicitly stated all of our assumptions in specific environments at the beginning of the paper, allowing readers to quickly verify the scope of our contributions. Extending our results to the non-convex setting and/or using stochastic gradients would be a very interesting contribution, but it would substantially change the scope of the current paper, which is already quite long. We consider either of those extensions to be exciting directions for future work in the field.
>
> - **Regarding non convexity**:
>
>   -- While a line of work has consisted in analyzing EF21 in non-convex settings [7–9], such analysis is not systematic; convexity assumptions are also common in the broader literature on distributed optimization, including [5, 6].
>
>   -- More importantly, results under weaker assumptions come with looser convergence guarantees, and we believe they do not always bring a better understanding of the methods: strong convexity allows us to show rates that are likely to be better characterizations of performance around local minima-which could be argued correspond better to training behavior ultimately observed in practice.
> Moreover, on the technical side, while extending our technique to PL condition is of great interest, we are unaware of tight _interpolation conditions_ for functions satisfying the PL condition. Providing them would on its own consitute a non-trivial research contribution. Overall, our choice is to focus on a setup powerful enough to provide a tight and fair comparison.
>
> ---
> We hope our clarifications help resolve the earlier concerns and would be grateful if you might reconsider the score. We’re glad to provide any further details you’d find useful.
>
> ---
> - [8] I. Fatkhullin, I. Sokolov, E. Gorbunov, Z. Li, P. Richtárik. *EF21 with bells & whistles: practical algorithmic extensions of modern error feedback*. JMLR , 2025.
> - [9] L. Condat, K. Yi, P. Richtárik. *EF-BV: A unified theory of error feedback and variance reduction mechanisms for biased and unbiased compression in distributed optimization*. Neurips, 2022.

---

> > ### Comment · Reviewer_f2XL · 2025-08-05
> >
> > Dear Authors,
> >
> > Thank you for the detailed reply!
> >
> >  >Practical relevance beyond distributed optimization
> >
> > I cannot fully agree with this point. In training with quantization, compression operators are applied not only to gradients but also to the model parameters (i.e., iterations). This makes the overall technique significantly different.
> > Moreover, it is not clear whether the concept of contractiveness of compressors is applicable in the context of quantization. It might be more relevant for sparsification.
> >
> > Overall, while quantization-aware training shares some conceptual similarities, it is substantially different from the setting considered here.
> >
> > >Numerous key papers derive their analysis in the single-node case
> >
> > I respectfully disagree with this argument. Papers [3–6] were published before 2020 and, while they are foundational and pioneering contributions to the field, they do not reflect the current direction of research. At the time of their publication, focusing on the single-node setting was appropriate, as the approach was still in its early stages and such simplifications helped build theoretical understanding.
> >
> > However, the field has since evolved significantly. Recent works increasingly address more realistic and practically relevant settings involving multiple nodes, which better capture the challenges encountered in modern applications.
> >
> > Furthermore, it is worth noting that paper [7] explicitly considers both single-node and multi-node cases, suggesting that the research community has been moving toward broader and more general settings.
> >
> > > Comparing EF, EF21, and CGD is far from straightforward even in the single-node case, and our analysis in the single-agent case already brings valuable insights
> >
> > I agree that the analysis is challenging even in the single-node case. However, I find it difficult to draw practical insights from this setting, as the behavior in the multi-node case can differ significantly from the single-node scenario. That said, I acknowledge that your contribution is solid from a theoretical perspective, I am simply trying to better understand the motivation behind focusing on the single-node setting.
> >
> > Additionally, I would like to point out that connections between EF and EF21 are discussed in the following paper:
> >
> > Gruntkowska, Kaja, Alexander Tyurin, and Peter Richtárik. "EF21-P and Friends: Improved Theoretical Communication Complexity for Distributed Optimization with Bidirectional Compression."* In International Conference on Machine Learning, pp. 11761–11807. PMLR, 2023.
> >
> > >Regarding non convexity:
> >
> > I mentioned this point to ensure there is no apparent contradiction between the fact that EF and EF21 achieve the same convergence rates in the strongly convex case, while EF21 appears to perform better in the nonconvex setting, as claimed by the authors of EF21.
> >
> > I look forward to hearing your response.
> >
> > Best regards,
> >
> > Reviewer

---

> > > ### Author Response · Authors · 2025-08-06
> > >
> > > We appreciate your feedback and would like to briefly address several of your points.
> > >
> > > >...In training with quantization, compression operators are applied not only to gradients but also to the model parameters (i.e., iterations)....
> > >
> > > Our aim is to demonstrate that single-agent results remain practically relevant beyond distributed optimization. Although existing methods in quantization-aware training differs in certain respects (something we agree on), these results can still offer useful insights as inspiration for future contributions in that field.
> > >
> > > Most importantly, EF already precisely appears in the sparse-neural-network literature: this alone seems to be sufficient evidence underscoring the practical relevance of single-agent results. The next draft will focus on this motivation.
> > >
> > > >...Papers [3–6] were published before 2020 and, while they are foundational and pioneering contributions to the field, they do not reflect the current direction of research...
> > >
> > > > Recent works increasingly address more realistic and practically relevant settings involving multiple nodes, which better capture the challenges encountered in modern applications.
> > >
> > > >Furthermore, it is worth noting that paper [7] explicitly considers both single-node and multi-node cases, suggesting that the research community has been moving toward broader and more general settings.
> > >
> > > We believe this is almost a philosophical debate:
> > > - Firstly, we argue that many "practical" optimization paradigms and methods (e.g., Adam, Adamax, etc.) are deeply rooted in the understanding of simpler setups. In particular, clean theoretical understanding of their roots (Nesterov, Heavy ball, coordinate methods, quasi-Newton methods, etc.) is arguably what allowed to naturally arrive at such practical algorithms.
> > > - In addition, we argue that one could build infinitely many variations around error feedback schemes (different iterative structures, different parameters, different guarantees, etc.). We believe that the community needs principles/techniques to be able to properly distinguish/choose between those methods and their properties. Again, we believe this is fundamental to allow the community to go further and build on those tools and their conclusions.
> > > - Finally, we would feel uncomfortable saying we understand an algorithmic technique without being able to perfectly grasp its subtleties in the simplest scenarios. In that sense, we believe that seeking to rank classical error feedback schemes in their simplest form (e.g., single-agent) is a sensible question.
> > >
> > > Of course, all of the points above are somewhat unquantifiable and perhaps subjective. But we would argue that the nature of those questions is naturally subjective. As a side note, we also feel there is a bit of subjectivity in asserting a trend. For instance, a subset of the authors cited by the reviewer still work on this exact topic in the single agent case (see e.g., "Error Feedback for Smooth and Nonsmooth Convex Optimization with Constant, Decreasing and Polyak Stepsizes", submitted to ICLR 2025).
> > >
> > > Based on those arguments, we would like to argue that our paper contributes grounding for the literature in two ways: (i) we provide new *optimal* Lyapunov functions, and (ii) we provide tight analyses for them. This means that we are absolutely sure about the ranking of the algorithms under consideration and their exact worst-case behavior in our setting. While we fully acknowledge recent trends toward multi-agent and stochastic settings, with the community moving toward more general settings,  our work formalizes settings that have already been studied. In our view, both lines of work are valuable, complementary, and necessary for the future.
> > >
> > > >Additionally, I would like to point out that connections between EF and EF21 are discussed in the
> > >
> > > We are open to including a discussion of this equivalence in the paper and would welcome any additional clarification you can provide. Our current understanding is that the equivalence pertains to EF21-P (described by the authors as the ‘primal’ of EF21), and that the equivalence holds only up to a permutation of sign, which is a variant of EF we do not evaluate in our paper.
> > >
> > > Lastly, we would like to highlight two points that we believe address the key concerns raised in your review:
> > > 1. **Assumption 1 is made for deterministic compressors:** As discussed, our analysis is indeed also valid for stochastic compressors
> > > 2. **Could the authors clarify any potential benefits of using such methods in the single-node case?:** We will add the explicit connection to sparse training of neural networks to highlight the direct practical relevance of the single-agent assumption
> > >
> > > Thank you again for your engagement in the discussion and constructive feedback. We hope this discussion has addressed your concerns and further highlighted the paper’s relevance in both theory and practice. If you find our response convincing, we would greatly appreciate an updated assessment.

---

> > > > ### Comment · Reviewer_f2XL · 2025-08-08
> > > >
> > > > Thank you very much for your thoughtful and detailed response.
> > > >
> > > > While I recognize that there are some similarities, I do not see a strong connection between quantization-aware training and error-feedback methods in the single-node setting, based on the points I raised earlier.
> > > >
> > > > I appreciate the rigorous analysis and the use of carefully constructed Lyapunov functions. However, I remain somewhat unconvinced that the insights obtained in the single-node case with deterministic gradients can be directly extended to more general or practical scenarios.
> > > >
> > > > As mentioned before, I believe the single-node setting with deterministic gradients is quite limited in scope. In my view, extending the analysis to the multi-node setting would be more impactful and important to consider.
> > > >
> > > > With all due respect to your work and perspective, I will keep my current score. Thank you again for the constructive and engaging discussion. I hope my comments are useful to you.
> > > >
> > > > Best regards,
> > > >
> > > > Reviewer

---

### Official Review · Reviewer_xTH1 · 2025-07-03

**Clarity:** 2
**Significance:** 3
**Originality:** 3
**Rating:** 4
**Confidence:** 3

**Summary:**

This work applies performance estimation framework (PEP) via SDP reformulation to convergence analysis of popular federated learning algorithms (CGD, EF and EF21) using contractive compression. While the analysis is restricted to smooth, strongly convex, single node (agent) setup where EF and EF21 are not helpful (meaning that CGD already converges and achieves tight rate), the use of PEP framework delivers some interesting insights. First, two new Lyapunov functions are found for EF and EF21, and contraction rates is shown to be equivalent despite different Lyapunov functions.

While I believe this work bring some useful insights for the study of compressed gradient methods, there are some formal inaccuracies in the Appendix and lack of explanation of some key steps. These issues summarized below are critical and have to be addressed in detail in the next revision before the paper is ready for publication. I am happy to increase my numerical rating if these concerns are addressed in the rebuttal.

**Questions:**

From analysis, CGD allows larger step-size of order $1/L$ than EF and EF21 which require $(1-\sqrt{\epsilon})/L$. This can be also seen from upper bounds and convergence analysis. Is there an intuitive explanation why this happens?

**Ethical Concerns:**

["NO or VERY MINOR ethics concerns only"]

**Final Justification:**

My main concern was about the SDP formulation under contractive compressors and the potential limitation that the study is limited to linear compressors. The authors explained some analysis steps in more detail and corrected some mistakes that made it difficult to understand some parts of the analysis. I am now mostly convinced that the paper indeed works with a general deterministic contractive compressor, and therefore increase my decision to "accept".

**Quality:**

2

**Strengths And Weaknesses:**

**Strengths:**

1. The paper applies a general methodology of performance estimation problem to a relevant and modern machine learning problem. The analysis yields interesting conclusions about two most popular error feedback schemes reinforcing their equivalence.

2. New tight Lyapunov functions are identified, which can potentially help to further improve the analysis of these algorithms in distributed settings and design even better algorithms.

**Weaknesses:**

1. Why interpolation conditions for the compression look as in equations (35,36) in supplementary material? This requires more detailed explanation as it is the key non-trivial extension of the prior work on PEP.

2. Since I don't fully understand the point (could be due to poor explanation), it seems to me that the analysis does not actually cover the entire class of contractive compressors. I am afraid that it is restricted to linear contractive compressors such as the one shown in tightness analysis in equation (50). If that is the case, then such analysis is very limited and not particularly insightful. Specifically, EF21 is equivalent to heavy ball momentum (if restricted to linear compressors), which has already been analysed using such PEP framework, and for such linear compressors EF = EF21 as shown in (Richtarik et al 2021), thus most claims in the paper follow immediately. In fact, from the practical viewpoint, the most interesting contractive compressor is Top-$K$, but I am not sure this is covered in this framework.

3. It is unclear why the resulting Lyapunov function for EF21 does not include the function value? I would expect it should since analysis in (Richtarik et al 2021) is conducted using the function value as the potential and is tight if I understand it correctly.


Typos and Mistakes:

In equation (21), $\nabla f(x_0)$ should be $f_0$.

In equation (22), it is unclear how to pick elements with $i = 1$ and $i = *$, because neither the gram matrix $G$ nor matrix $B$ contain elements with $i = 1$.

If I understand it correctly, the definition of large $C_{i,j}$ coefficients in line 591 can be simplified using a norm squared instead of inner products.

On line 595 there is likely a wrong pointer to equation (27) which wasn't discussed at this point yet.

I assume it is just a typo, but coefficients $C_i$ in Lemma 1 should have a superscript EF. Same issue for analogous lemma for EF21. Moreover there is an issue with superscript in Lemma 2 in the SDP.

---

> ### Author Rebuttal · Authors · 2025-07-30
>
> We sincerely thank the reviewer for their detailed and thoughtful comments, particularly regarding the SDP formulation in the appendix. We appreciate the reviewer’s careful reading, as some highlighted questions will help significantly improve the clarity of the manuscript.
>
> We first want to emphasize that the derivations and theoretical results presented in the main text remain correct and unaffected. In particular, it should be noted that the proofs of Theorems 1 and 2 do not rely on the SDP formulation and can be verified independently. We hope this reassures the reviewer regarding the validity of our results.
>
> Below, we respond point by point to the reviewer’s remarks.
>
> >Why interpolation conditions for the compression look as in equations (35,36) in supplementary material?
>
> Thank you for pointing this out. First, we acknowledge a clear typo in equation (36):
> $$
> C\_i^{EF} = \\begin{bmatrix} \\bar{g}\_i \\\\ \\bar{c}\_i \\\\ \\mathbf{\\bar{c}}\_i \\end{bmatrix}^\\top C\_{EF} \\begin{bmatrix}\\bar{g}\_i \\\\ \\bar{c}\_i \\\\ \\mathbf{\\bar c}\_i\\end{bmatrix}.
> $$
> This should instead be
> $$
> C\_i^{EF} = \\begin{bmatrix}\\bar{g}\_i \\\\ \\bar{c}_i \\\\ \\mathbf{\\bar{e}}\_i\\end{bmatrix}^\\top C\_{EF} \\begin{bmatrix}\\bar{g}\_i \\\\ \\bar{c}\_i \\\\ \\mathbf{\\bar e}\_i\\end{bmatrix}.
> $$
>
> We apologize for this oversight, which may have made interpretation difficult.
>
> Let us now explain why we define equation 36 as we did. The interpolation condition, for any $i \\in \\{0, 1\\}$  can be written as
> $$
> ||(\eta g_i + e_i) - c_i|| ^2 \leq \epsilon ||\eta g_i + e_i||^2,
> $$
> which is a direct discretization of Assumption 1 or equivalently,
> $$
> ||(\eta g_i + e_i) - c_i|| ^2 - \epsilon ||\eta g_i + e_i||^2 \le 0.
> $$
>
> Equation 36 is simply a rephrasing of this condition, after expansion, in terms of the basis vectors we defined:
> $$
> \\begin{aligned}
> &||(\\eta g\_i + e\_i) - c\_i|| ^2 - \\epsilon ||\\eta g\_i + e\_i||^2  \\\\
> &=||c\_i||^2 + \\eta^2 ||g\_i||^2 + ||e\_i|| ^2 - 2\\eta \\langle g\_i, c\_i \\rangle -2 \\eta \\langle e\_i, g\_i \\rangle + 2 \\langle c\_i , e\_i \\rangle  - \\epsilon \\eta^2 || g\_i||^2 - \\epsilon ||e\_i||^2 -  2\\epsilon \\eta \\langle g\_i, c\_i \\rangle \\\\
> &= \\text{Tr}\\left(\\begin{bmatrix}g\_i \\\\ c\_i \\\\ e\_i\\end{bmatrix}^\\top C\^{EF} \\begin{bmatrix}g\_i \\\\ c\_i \\\\ e\_i\\end{bmatrix}\\right).
> \\end{aligned}
> $$
> This can be seen directly by matching the coefficients we now wrote down explicitly, to that of the matrix in equation (35).
> Ultimately, $C_i^{EF}$ is defined such that the outer basis vectors are "picking" the contributions from the right vectors, and then assigning them the weights given in equation 35, i.e. s.t.:
> $$
> \\begin{aligned}
>  \\text{Tr}\\left(\\begin{bmatrix}g\_i \\\\ c\_i \\\\ e\_i\\end{bmatrix}^\\top C\^{EF} \\begin{bmatrix}g\_i \\\\ c\_i \\\\ e\_i\\end{bmatrix}\\right)
> \\end{aligned} = \\begin{aligned}
>  \\text{Tr}\\left(  B^\\top C_i\^{EF}  B\\right)
> \\end{aligned}
> $$
> From there on, the derivation follows step 1 and 2 from section A. The primal constraint  $\\text{Tr}\\left(  B^\\top C_i\^{EF}  B\\right) = \\text{Tr}\\left( C_i\^{EF}  G \\right) \le 0 $ leads to the term $\sum_{i \in\{0,1\}} \gamma_i \cdot C_i^{E F}$ in eq. (EF-SDP) (following eq. (36)).
>
>
> We will provide the details of this computation in the next version.
>
> Additionally, we acknowledge that in the same subsection, we reused the notation $C_{ij}$, $c_{ij}$ (later referred to as $M_{ij}$ and $m_{ij}$) for the smoothness and strong convexity interpolation conditions. This overlaps with the notation used for $C_i^{EF}$ and $C_i^{EF21}$, and we understand this may have contributed to confusion. We sincerely apologize and have revised the appendix to improve the clarity and consistency of notation throughout.
>
> >Since I don't fully understand the point (could be due to poor explanation), it seems to me that the analysis does not actually cover the entire class of contractive compressors. I am afraid that it is restricted to linear contractive compressors such as the one shown in tightness analysis in equation (50).  [...]. In fact, from the practical viewpoint, the most interesting contractive compressor is Top-$K$, but I am not sure this is covered in this framework.
>
> We hope the explanation provided above clarifies how Assumption 1 is exactly encoded in the SDP without restriction. To further reassure the reviewer, we emphasize that the analysis of convergence rates is entirely decoupled from the SDP—it directly invokes Assumption 1 in the proofs, and does not necessitate any linearity. The SDP was used only to design the Lyapunov function we use in our analysis.
>
> Regarding equation (50) specifically: it appears in the **lower-bound** construction, which is separate from the upper bound. We obtain this lower bound using a linear compressor given by eq. (50), on a specific function. It is sufficient to achieve the lower bound on this linear compressor to conclude, as it belongs to the more general set of compressors. The upper bound is obtained under the full generality of Assumption 1, and thus beyond linear contractiveness.
>
> The Top-$K$ compressor is thus indeed included in our analysis.
>
> >It is unclear why the resulting Lyapunov function for EF21 does not include the function value? I would expect it should since analysis in (Richtarik et al 2021) is conducted using the function value as the potential and is tight if I understand it correctly.
>
> It is indeed an interesting consequence of our analysis that the optimal Lyapunov function does not correspond to the ones in  the literature (even for EF!). This is actually one strength of our approach to be able to automatically and exactly derive the best possible Lyapunov functions.
>
> Regarding the analysis of (Richtarik et al 2021), there are a couple of differences on the assumption set. We are unsure about which exact tightness result you mention. Regardless, tightness results **for a given Lyapunov function** are very distinct from tightness results that say that **no** Lyapunov function can provide a better rate (and our result is tight in both senses). Getting results of the latter type typically requires using performance estimation techniques. It is thus not surprising that the use of these advanced tools enables us to find better Lyapunov functions.
>
> We thank the reviewer for pointing out typos, which will be remedied in the next revision. We would just like to comment on the following:
>
> >If I understand it correctly, the definition of large $C_{i, j}$ coefficients in line 591 can be simplified using a norm squared instead of inner products.
>
> Note that the basis vectors are row vectors, expressions like $\bar{g}_i^\top \bar{x}_j$ denote outer products (i.e. matrices), and not inner products (scalar). While we could reformulate these terms using a custom norm operator representing self-outer-products, we prefer to keep the current notation to avoid introducing new definitions that would require additional explanation. Instead, we have emphasized in the revised draft that the basis vectors are row vectors, to prevent further confusion.
>
> > From analysis, CGD allows larger step-size of order $1/L$ than EF and EF21 which require $(1-\sqrt{\epsilon})/L$. This can be also seen from upper bounds and convergence analysis. Is there an intuitive explanation why this happens?
>
> We note that the optimal step size for CGD is given by [2]
> $$
> \eta = \frac{2}{(1 - \sqrt{\epsilon})\mu + (1 + \sqrt{\epsilon}) L}.
> $$
> The optimal rate for CGD is also given by [1, 2]
> $$
> \left(\frac{\kappa_{\sqrt{\epsilon}} - 1}{\kappa_{\sqrt{\epsilon}} + 1} \right)^2,
> $$
> where $\kappa_{\sqrt{\epsilon}} = \frac{L (1 + \sqrt \epsilon)}{\mu (1 - \sqrt \epsilon)}$ (note that we define the conditioning number inversely compared to [1]). An intuition for this is essentially that the uncertainty in the update direction is "rescaling" $L$ by adding "$\sqrt{\epsilon}$ uncertainty", and doing the converse operation for $\mu$.
>
> As far as our step size is concerned, it is given by
> $$
> \eta = \left(\frac{2}{L + \mu}\right) \left(\frac{1 - \sqrt{\epsilon}}{1 + \sqrt{\epsilon}}\right).
> $$
> That is to say: in CGD, the "rescaling" is applied at the level of the smoothness and strong convexity constants, whereas the scaling for EF/EF21 is applied at the level of the step size.
>
> ---
>
> We hope our responses have clarified any misunderstandings and that you will consider revising your score accordingly. We would be happy to provide further clarification if needed.
>
> ---
> References:
> - [1] E. De Klerk, F. Glineur, A. Taylor. _On the worst-case complexity of the gradient method with exact line search for smooth strongly convex functions_. Optimization Letters, 2016.
> - [2] O. Gannot. *A frequency-domain analysis of inexact gradient methods*. Mathematical programming, 2020.

---

> > ### Comment · Reviewer_xTH1 · 2025-08-05
> >
> > I thank the authors for addressing points 1 and 2. After checking this, I have reconsidered my evaluation. Regarding point 3, it would be interesting to understand if there are any implications of this fact that the Lyapunov function using the function value is not optimal in this sense. E.g., does it mean it is always better to use a different analysis (coming from this PEP formulation) instead of a standard function value-based analysis (when possible)?

---

> > > ### Author Response · Authors · 2025-08-06
> > >
> > > Thank you! Regarding any wider implications: we are not aware of any consistent rules to say whether functional residual is optimal or not across different problems. To further explore this question, we went ahead and checked a more flexible version of (Richtarik et al 2021)’s Lyapunov function, where we allow for any weighting between the functional residual and the norm distance between the gradient and communicated message (which strictly includes the one used in Theorem 2 in (Richtarik et al 2021)). For any step size $\eta$ and contraction factor $\epsilon$, we optimized the weighting to get the best possible contraction rate numerically. We found that this family of function-value-based Lyapunov functions leads to numerically worse contraction rates than the one we get using the PEP framework of the article, for *any* step size $\eta$ and *any* $\epsilon$, which is a numerical proof that those Lyapunov functions are always suboptimal under our set of assumptions.
> > >
> > > We will include in the paper a discussion on this topic and the corresponding plots.
> > >
> > > Thank you again for your feedback and support!

---

### Note · Authors · 2025-08-16

We sincerely thank the reviewers and the AC for their careful engagement, constructive feedback, and the opportunity to strengthen our work.

Across the reviews, several strengths were consistently emphasized: clarity and accessibility of the writing (f2XL, sQ6n, z1Gw), rigor and originality of the theoretical analysis (xTH1, z1Gw), and the quality of figures, tables, and proofs (f2XL, sQ6n). Reviewer xTH1 highlighted the novelty of deriving new tight Lyapunov functions, while Reviewer z1Gw stressed that the equivalence between EF and EF21's rates was meaningful.

The discussion also allowed us to clarify key points. Among other things, we confirmed that our results extend beyond deterministic compressors to stochastic contractive compressors, and we will add the corresponding assumption and theorem statements. We numerically demonstrated that function-value-based Lyapunov functions are always suboptimal under our assumptions, strengthening our main contribution. We clarified optimal stepsizes and rates for CGD, EF, and EF21.

Remaining debate focused on scope. Two reviewers (f2XL, z1Gw) initially questioned the practical relevance of the single-agent case. We respectfully emphasize our strong belief that tight analyses in simple regimes are an essential step toward understanding more complex ones. Without solid grounding in such cases, development in distributed, stochastic, or non-convex settings risks being inconclusive. We agree that both simple and advanced setups deserve attention, and we see our work as contributing to this broader balance. Moreover, a side remark is that single agent EF is also used for sparse NN. Importantly, even reviewers who raised this concern acknowledged the theoretical value of our work, and in the case of z1Gw, explicitly supported acceptation after the discussion.

For the final version, we plan to: (i) revise the abstract, stating the single-agent focus explicitly; (ii) add stochastic compressor results; (iii) include explicit rate formulas and comparisons; (iv) expand discussion of EF vs EF21, and EF21-P; (v) fix typos, and add content on practical relevance and limitations, (vi) add function-value-Lyapunov numerical results.

In summary, our key contribution is the first tight analysis of EF and EF21, yielding new optimal Lyapunov functions, matching lower bounds, and sharp performance guarantees. We thank the reviewers and AC again for the constructive process, which will significantly strengthen the final version

---

### Decision · Program_Chairs · 2025-09-17

**Decision:**

Accept (poster)

**Comment:**

This paper provides a tight theoretical analysis of EF, EF21, and compressed gradient descent in the simplified single-node smooth and strongly convex setting. Using the performance estimation framework, the authors derive optimal Lyapunov functions and matching lower bounds, establishing sharp convergence guarantees. The results show that EF and EF21 achieve equivalent rates despite different Lyapunov structures, and they clarify the performance trade-offs with compressed gradient descent under different stepsize regimes.

The reviewers praised the clarity and rigor of the analysis and the novelty of the Lyapunov constructions. The main concern was scope, as the work is limited to the single-node deterministic case. The authors clarified the extension to stochastic compressors and added explicit rate formulas. While one reviewer remained skeptical about practical impact, others found the responses convincing, and the discussion shifted the balance toward acceptance.

In my view, the work makes a solid theoretical contribution. The optimal Lyapunov functions are insightful: for EF, the form aligns with earlier analyses but gives exact weighting, while for EF21 the analysis reveals a structure not considered before. Although the single-node focus is a limitation, the results provide useful clarity. For the final version, the authors should make the changes promised in the final remarks, especially clarifying the scope and adding the key references mentioned in the rebuttal, while using their discretion if some works are too distant.